# KNOWLEDGE DISTILLATION AS SEMIPARAMETRIC INFERENCE

**Tri Dao**[1]**, Govinda M. Kamath**[2]**, Vasilis Syrgkanis**[2]**, Lester Mackey**[2]
[1] Department of Computer Science, Stanford University
[2] Microsoft Research, New England
`trid@stanford.edu,{govinda.kamath,vasy,lmackey}@microsoft.com`

## ABSTRACT

A popular approach to model compression is to train an inexpensive student model to mimic the class probabilities of a highly accurate but cumbersome teacher model. Surprisingly, this two-step knowledge distillation process often leads to higher accuracy than training the student directly on labeled data. To explain and enhance this phenomenon, we cast knowledge distillation as a semiparametric inference problem with the optimal student model as the target, the unknown Bayes class probabilities as nuisance, and the teacher probabilities as a plug-in nuisance estimate. By adapting modern semiparametric tools, we derive new guarantees for the prediction error of standard distillation and develop two enhancements—cross-fitting and loss correction—to mitigate the impact of teacher overfitting and underfitting on student performance. We validate our findings empirically on both tabular and image data and observe consistent improvements from our knowledge distillation enhancements.

## 1 INTRODUCTION

Knowledge distillation (KD) (Craven & Shavlik, 1996; Breiman & Shang, 1996; Bucila et al., 2006; Li et al., 2014; Ba & Caruana, 2014; Hinton et al., 2015) is a widely used model compression technique that enables the deployment of highly accurate predictive models on devices such as phones, watches, and virtual assistants (Stock et al., 2020). KD operates by training a compressed student model to mimic the predicted class probabilities of an expensive, high-quality teacher model. Remarkably and across a wide variety of domains (Hinton et al., 2015; Sanh et al., 2019; Jiao et al., 2019; Liu et al., 2018; Tan et al., 2018; Fakoor et al., 2020), this two-step process often leads to higher accuracy than training the student directly on the raw labeled dataset.

While the practice of KD is now well developed, a general theoretical understanding of its successes and failures is still lacking. As we detail below, a number of authors have argued that the success of KD lies in the more precise "soft labels" provided by the teacher's predicted class probabilities. Recently, Menon et al. (2020) observed that these teacher probabilities can serve as a proxy for the *Bayes probabilities* (i.e., the true class probabilities) and that the closer the teacher and Bayes probabilities, the better the student's performance should be.

Building on this observation, we cast KD as a plug-in approach to *semiparametric inference* (Kosorok, 2007): that is, we view KD as fitting a student model $\hat{f}$ in the presence of nuisance (the Bayes probabilities $p_0$) with the teacher's probabilities $\hat{p}$ as a plug-in estimate of $p_0$. This insight allows us to adapt modern tools from semiparametric inference to analyze the error of a distilled student in Sec. 3. Our analysis also reveals two distinct failure modes of KD: one due to teacher overfitting and data reuse and the other due to teacher underfitting from model misspecification or insufficient training. In Sec. 4, we introduce and analyze two complementary KD enhancements that correct for these failures: *cross-fitting*—a popular technique from semiparametric inference (see, e.g., Chernozhukov et al., 2018)—mitigates teacher overfitting through data partitioning while *loss correction* mitigates teacher underfitting by reducing the bias of the plug-in estimate $\hat{p}$. The latter enhancement was inspired by the *orthogonal machine learning* (Chernozhukov et al., 2018; Foster & Syrgkanis, 2019) approach to semiparametric inference which suggests a particular adjustment for the teacher's log probabilities. We argue in Sec. 4 that this orthogonal correction minimizes the teacher bias but often at the cost of

unacceptably large variance. Our proposed correction avoids this variance explosion by balancing the bias and variance terms in our generalization bounds.

In Sec. 5, we complement our theoretical analysis with a pair of experiments demonstrating the value of our enhancements on six real classification problems. On five real tabular datasets, cross-fitting and loss correction improve student performance by up to 4% AUC over vanilla KD. Furthermore, on CIFAR-10 (Krizhevsky & Hinton, 2009), a benchmark image classification dataset, our enhancements improve vanilla KD accuracy by up to 1.5% when the teacher model overfits.

**Related work.** Since we cannot review the vast literature on KD in its entirety, we point the interested reader to Gou et al. (2020) for a recent overview of the field. We devote this section to reviewing theoretical advances in the understanding of KD and summarize complementary empirical studies and applications of in the extended literature review in App. A.

A number of papers have argued that the availability of soft class probabilities from the teacher rather than hard labels enables us to improve training of the student model. This was hypothesized in Hinton et al. (2015) with empirical justification. Phuong & Lampert (2019) consider the case in which the teacher is a fixed linear classifier and the student is either a linear model or a deep linear network. They show that the student can learn the teacher perfectly if the number of training examples exceeds the ambient dimension. Vapnik & Izmailov (2015) discuss the setting of learning with privileged information where one has additional information at training time which is not available at test time. Lopez-Paz et al. (2015) draw a connection between this and KD, arguing that KD is effective because the teacher learns a better representation allowing the student to learn at a faster rate. They hypothesize that a teacher's class probabilities enable student improvement by indicating how difficult each point is to classify. Tang et al. (2020) argue using empirical evidence that label smoothing and reweighting of training examples using the teacher's predictions are key to the success of KD. Mobahi et al. (2020) analyzed the case of self-distillation in which the student and teacher function classes are identical. Focusing on kernel ridge regression models, they proved that self-distillation can act as increased regularization strength. Bu et al. (2020) considers more generic model compression in a rate-distortion framework, where the rate is the size of the student model and distortion is the difference in excess risk between the teacher and the student. Menon et al. (2020) consider the case of losses such that the population risk is linear in the Bayes class probabilities. They consider *distilled empirical risk* and *Bayes distilled empirical risk* which are the risk computed using the teacher class probabilities and Bayes class probabilities respectively rather than the observed label. They show that the variance of the Bayes distilled empirical risk is lower than the empirical risk. Then using analysis from Maurer & Pontil (2009); Bennett (1962), they derive the excess risk of the distilled empirical risk as a function of the $\ell_2$ distance between the teacher's class probabilities and the Bayes class probabilities. We significantly depart from Menon et al. (2020) in multiple ways: i) our Thm. 1 allows for the common practice of data re-use, ii) our results cover the standard KD losses SEL and ACE which are non-linear in $p_0$, iii) we use localized Rademacher analysis to achieve tight fast rates for standard KD losses, and iv) we use techniques from semiparametric inference to improve upon vanilla KD.

## 2 KNOWLEDGE DISTILLATION BACKGROUND

We consider a multiclass classification problem with $k$ classes and $n$ training datapoints $z_i = (x_i, y_i)$ sampled independently from some distribution $\mathbb{P}$. Each feature vector $x$ belongs to a set $\mathcal{X}$, each label vector $y \in \{e_1, ..., e_k\} \subset \{0,1\}^k$ is a one-hot encoding of the class label, and the conditional probability of observing each label is the *Bayes class probability* function $p_0(x) = \mathbb{E}[Y \mid X = x]$. Our aim is to identify a scoring rule $f : \mathcal{X} \to \mathbb{R}^k$ that minimizes a prediction loss on average under the distribution $\mathbb{P}$.

**Knowledge distillation.** Knowledge distillation (KD) is a two-step training process where one first uses a labeled dataset to train a teacher model and then trains a student model to predict the teacher's predicted class probabilities. Typically the teacher model is larger and more cumbersome, while the student is smaller and more efficient. Knowledge distillation was first motivated by model compression (Bucila et al., 2006), to find compact yet high-performing models to be deployed (such as on mobile devices).

In training the student to match the teacher's prediction probability, there are several types of loss functions that are commonly used. Let $\hat{p}(x) \in \mathbb{R}^k$ be the teacher's vector of predicted class probabilities, $f(x) \in \mathbb{R}^k$ be the student model's output, and $[k] \triangleq \{1, 2, ..., k\}$. The most popular distillation loss

functions[1] $\ell(z;f(x),\hat{p}(x))$ include the squared error logit (SEL) loss (Ba & Caruana, 2014)

$$\ell_{\mathrm{se}}(z;f(x),\hat{p}(x)) \triangleq \sum_{j\in[k]} \tfrac{1}{2}(f_j(x)-\log(\hat{p}_j(x)))^2 \qquad \text{(SEL)}$$

and the annealed cross-entropy (ACE) loss (Hinton et al., 2015)

$$\ell_\beta(z;f(x),\hat{p}(x)) = -\sum_{j\in[k]} \frac{\hat{p}_j(x)^\beta}{\sum_{l\in[k]}\hat{p}_l(x)^\beta} \log\left(\frac{\exp(\beta f_j(x))}{\sum_{l\in[k]}\exp(\beta f_l(x))}\right) \qquad \text{(ACE)}$$

for an inverse temperature $\beta > 0$. These loss functions measure the divergence between the probabilities predicted by the teacher and the student.

A student model trained with knowledge distillation often performs better than the same model trained from scratch (Bucila et al., 2006; Hinton et al., 2015). In Secs. 3 and 4, we will adapt modern tools from semiparametric inference to understand and enhance this phenomenon.

## 3 DISTILLATION AS SEMIPARAMETRIC INFERENCE

In semiparametric inference (Kosorok, 2007), one aims to estimate a target parameter or function $f_0$, but that estimation depends on an auxiliary *nuisance function* $p_0$ that is unknown and not of primary interest. We cast the knowledge distillation process as a semiparametric inference problem, by treating the unknown Bayes class probabilities $p_0$ as nuisance and the teacher's predicted probabilities as a plug-in estimate of that nuisance. This perspective allows us bound the generalization of the student in terms of the mean squared error (MSE) between the teacher and the Bayes probabilities. In the next section (Sec. 4) we use techniques from semiparametric inference to enhance the performance of the student. The interested reader could consult Tsiatis (2007) for more details on semiparametric inference.

Our analysis starts from taking the following perspective on distillation. For a given pointwise loss function $\ell(z;f(x),p_0(x))$, we view the goal of the student as minimizing an oracle population loss over a function class $\mathcal{F}$,

$$L_D(f,p_0) = \mathbb{E}[\ell(Z;f(X),p_0(X))] \quad \text{with} \quad f_0 \triangleq \operatorname{argmin}_{f\in\mathcal{F}} L_D(f,p_0).$$

The main hurdle is that this is objective depends on the unknown Bayes probabilities $p_0$. We view the teacher's model $\hat{p}$ as an approximate version of $p_0$ and bound the distillation error of the student as a function of the teacher's estimation error.

Typical semiparametric inference considers cases where $f_0$ is a finite dimensional parameter; however recent work of Foster & Syrgkanis (2019) extends this framework to infinite dimensional models $f_0$ and to develop statistical learning theory with a nuisance component framework. The distillation problem fits exactly into this setup.

**Bounds on vanilla KD** As a first step we derive a vanilla bound on the error of the distilled student model without any further modifications of the distillation process, i.e., we assume that the student is trained on the same data as the teacher and is trained by running empirical risk minimization (ERM) on the plug-in loss, plugging in the teacher's model instead of $p_0$, i.e.,

$$\hat{f} = \operatorname{argmin}_{f\in\mathcal{F}} L_n(f,\hat{p}) \quad \text{for} \quad L_n(f,\hat{p}) \triangleq \mathbb{E}_n[\ell(Z;f(X),\hat{p}(X))] \qquad \text{(Vanilla KD)}$$

where $\mathbb{E}_n[X] = \frac{1}{n}\sum_{i=1}^n X_i$ denotes the empirical expectation of a random variable.

**Technical definitions** Before presenting our main theorem we introduce some technical notation. For a vector valued function $f$ that takes as input a random variable $X$, we use the shorthand notation $\|f\|_{p,q} \triangleq \big\|\|f(X)\|_p\big\|_{L^q} = \mathbb{E}\big[\|f(X)\|_p^q\big]^{1/q}$. Let $\nabla_\phi$ and $\nabla_\pi$ denote the partial derivatives of $\ell(z;\phi,\pi)$, with respect to its second and third input correspondingly and $\nabla_{\phi\pi}$ the Jacobian of cross partial derivatives, i.e., $[\nabla_{\phi\pi}\ell(z;\phi,\pi)]_{i,j} = \frac{\partial^2}{\partial\phi_j\partial\pi_i}\ell(z;\phi,\pi)$. Finally, let

$$q_{f,p}(x) = \mathbb{E}[\nabla_{\phi\pi}\ell(Z;f(X),p(X))\,|\,X=x] \quad \text{and} \quad \gamma_{f,p}(x) = \mathbb{E}_{U\sim\mathrm{Unif}([0,1])}[q_{f,Up+(1-U)p_0}(x)].$$

**Critical radius** Finally, we need to define the notion of the critical radius (see, e.g., Wainwright (2019, 14.1.1)) of a function class, which typically provides tight learning rates for statistical learning theory tasks. For any function class $\mathcal{F}$ we define the localized Rademacher complexity as:

$$\mathcal{R}(\delta;\mathcal{F}) = \mathbb{E}_{X_{1:n},\epsilon_{1:n}}[\sup_{f\in\mathcal{F}:\|f\|_2\leq\delta}\tfrac{1}{n}\sum_{i=1}^n\epsilon_i f(X_i)]$$

where $\epsilon_i$ are i.i.d. random variables taking values equiprobably in $\{-1,1\}$. The *critical radius* of a class $\mathcal{F}$, taking values in $[-H,H]$, is the smallest positive solution $\delta_n$ to the inequality $\mathcal{R}(\delta;\mathcal{F}) \leq \frac{\delta^2}{H}$.

---

[1]These loss functions do not depend on the ground-truth label $y$, but we use the augmented notation $\ell(z;f(x),\hat{p}(x))$ to accommodate the enhanced distillation losses presented in Sec. 4.

**Theorem 1** (Vanilla KD analysis). *Suppose $f_0$ belongs to a convex set $\mathcal{F}$ satisfying the $\ell_2/\ell_4$ ratio condition $\sup_{f \in \mathcal{F}} \|f - f_0\|_{2,4}/\|f - f_0\|_{2,2} \leq C$ and that the teacher estimates $\hat{p} \in \mathcal{P}$ from the same dataset used to train the student. Let $\delta_{n,\zeta} = \delta_n + c_0 \sqrt{\frac{\log(c_1/\zeta)}{n}}$ for universal constants $c_0, c_1$ and $\delta_n$ an upper bound on the critical radius of the function class*
$$\mathcal{G} \triangleq \{z \rightarrow r(\ell(z; f(x), p(x)) - \ell(z; f_0(x), p(x))) : f \in \mathcal{F}, p \in \mathcal{P}, r \in [0,1]\}.$$
*Let $\mu(z) = \sup_\phi \|\nabla_\phi \ell(z; \phi, \hat{p}(x))\|_2$, and assume that the loss $\ell(z; \phi, \pi)$ is $\sigma$-strongly convex in $\phi$ for each $z$ and that each $g \in \mathcal{G}$ is uniformly bounded in $[-H, H]$. Then the Vanilla KD $\hat{f}$ satisfies*
$$\|\hat{f} - f_0\|_{2,2}^2 = \tfrac{1}{\sigma^2} O(\delta_{n,\zeta}^2 C^2 H^2 \|\mu\|_4^2 + \|\gamma_{f_0, \hat{p}}^\top (\hat{p} - p_0)\|_{2,2}^2) \quad \textit{with probability at least} \quad 1 - \zeta.$$

Thm. 1, proved in App. C, shows that vanilla distillation yields an accurate student whenever the teacher generalizes well (i.e., $\|\hat{p} - p_0\|_{2,2}$ is small) and the student and teacher model classes $\mathcal{F}$ and $\mathcal{P}$ are not too complex. The $\ell_2/\ell_4$ ratio requirement can be removed at the expense of replacing $\|\mu\|_4$ by $\|\mu\|_\infty = \sup_z |\mu(z)|$ in the final bound. Moreover, we highlight that the strong convexity requirement for $\ell$ is satisfied by all standard distillation objectives including SEL and ACE, as it is strong convexity with respect to the output of $f$ and not the parameters of $f$. Even this requirement could be removed, but this would yield slow rate bounds of the form: $\|\hat{f} - f_0\|_{2,2}^2 = O(\delta_{n,\zeta} + \|\gamma_{f_0, \hat{p}}^\top (\hat{p} - p_0)\|_{2,2}^2)$.

**Failure modes of vanilla KD** Thm. 1 also hints at two distinct ways in which vanilla distillation could fail. First, since the student only learns from the teacher and does not have access to the original labels, we would expect the student to be erroneous when the teacher probabilities are inaccurate due to model misspecification, an overly restrictive teacher function class, or insufficient training. Prop. 2, proved in App. D, confirms that, in the worst case, student error suffers from inaccuracy due to this *teacher underfitting* even when both the student and teacher belong to low complexity model classes.

**Proposition 2** (Impact of teacher underfitting on vanilla KD). *There exists a classification problem in which the following properties all hold simultaneously with high probability for $f_0 = \log(p_0)$:*

- *The teacher learns $\hat{p}(x) = \frac{1}{n(1+\lambda)} \sum_{i=1}^n y_i$ for all $x \in \mathcal{X}$ via ridge regression with $\lambda = \Theta(1/n^{1/4})$.*
- *Vanilla KD with SEL loss and constant $\hat{f}$ satisfies $\|\hat{f} - f_0\|_{2,2}^2 \geq \|\gamma_{f_0, p_0}^\top (\hat{p} - p_0)\|_{2,2}^2 = \Omega(\frac{1}{\sqrt{n}})$, matching the dependence of the Thm. 1 upper bound up to a constant factor.*
- *Enhanced KD with SEL loss, $\hat{\gamma}^{(t)} = \text{diag}(\frac{1}{\hat{p}^{(t)}})$, and constant $\hat{f}$ satisfies $\|\hat{f} - f_0\|_{2,2}^2 = O(\frac{1}{n})$.*

Second, the critical radius in Thm. 1 depends on the complexity of the teacher model class $\mathcal{P}$. If $\mathcal{P}$ has a large critical radius, then the student error bound suffers due to potential teacher overfitting *even if the teacher generalizes well*. Prop. 3, proved in App. E, shows that, in the worst case, this *teacher overfitting* penalty is unavoidable and does in fact lead to increased student error. This occurs as the student only has access to the teacher's training set probabilities which, due to overfitting, need not reflect its test set probabilities.

**Proposition 3** (Impact of teacher overfitting on vanilla KD). *There exists a classification problem in which the following properties all hold simultaneously with high probability for $f_0 = \mathbb{E}[\log(p_0(X))]$:*

- *The critical radius $\delta_n$ of the teacher-student function class $\mathcal{G}$ in Thm. 1 is a non-vanishing constant, due to the complexity of the teacher's function class.*
- *The Vanilla KD error $\|\hat{f} - f_0\|_{2,2}^2$ for constant $\hat{f}$ with SEL loss is lower bounded by a non-vanishing constant, matching the $\delta_n$ dependence of the Thm. 1 upper bound up to a constant factor.*
- *Enhanced KD with SEL loss, $\hat{\gamma}^{(t)} = 0$, and constant $\hat{f}$ satisfies $\|\hat{f} - f_0\|_{2,2}^2 = O(n^{-4/(4+d)})$.*

These examples serve to lower bound student performance in the worst case by the teacher's critical radius and class probability MSE, matching the upper bounds given in Thm. 1. However, we note that in other better-case scenarios vanilla distillation can perform better than the upper-bounding Thm. 1 would imply. In the next section, we adapt and generalize techniques from semiparametric inference to mitigate the effects of teacher overfitting and underfitting in all cases.

## 4 Enhancing Knowledge Distillation

To address the two distinct inefficiencies of vanilla distillation revealed in Sec. 3, we will adapt and generalize two distinct techniques from semiparametric inference: orthogonal correction and cross-fitting.

## 4.1 COMBATING TEACHER UNDERFITTING WITH LOSS CORRECTION

We can view the plug-in distillation loss $\ell(z;f(x),\hat{p}(x))$ as a zeroth order Taylor approximation to the ideal loss $\ell(z;f(x),p_0(x))$ around $\hat{p}$. An ideal first-order approximation would take the form

$$\ell(z;f(x),\hat{p}(x))+\langle p_0(x)-\hat{p}(x),\nabla_\pi\ell(z;f(x),\hat{p}(x))\rangle.$$

However, its computation also requires knowledge of $p_0$. Nevertheless, since $p_0(x)=\mathbb{E}[Y\,|\,X=x]$, we can always construct an unbiased estimate of the ideal first order term by replacing $p_0(x)$ with $y$:

$$\ell_{ortho}(z;f(x),\hat{p}(x))=\ell(z;f(x),\hat{p}(x))+\langle y-\hat{p}(x),\mathbb{E}[\nabla_\pi\ell(z;f(x),\hat{p}(x))\,|\,x]\rangle. \tag{1}$$

For standard distillation base losses like SEL and ACE, the *orthogonal loss* (1) has an especially simple form, as $\nabla_\pi\ell(z;f(x),\hat{p}(x))$ is linear in $f$. Indeed, this is true more generally for the following class of Bregman divergence losses.

**Definition 1** (Bregman divergence losses). *Any* Bregman divergence *loss function of the form*

$$\ell(z;f(x),p(x))\triangleq\Psi(f(x))-\Psi(g(p(x)))-\langle\nabla_g\Psi(g(p(x))),f(x)-g(p(x))\rangle \quad has$$

$$\ell_{ortho}(z;f(x),p(x))=\ell(z;f(x),p(x))+(y-p(x))^\top\nabla_p g(p(x))^\top\nabla_{gg}^2\Psi(g(p(x)))f(x)+\mathrm{const} \tag{2}$$

*with the second term bilinear in $f(x)$ and $y-p(x)$. For the SEL loss, $\Psi(s)=\frac{1}{2}\|s\|_2^2$, $g(p)=\log(p)$, and the correction matrix $\nabla_p g(p(x))^\top\nabla_{gg}^2\Psi(g(p(x)))=\mathrm{diag}(\frac{1}{p(x)})$. Similarly, the ACE loss falls into the class of Bregman divergence losses.*

We will show that orthogonal correction (1) can significantly improve student bias due to teacher underfitting; however, for our standard distillation losses (SEL and ACE), the same orthogonal correction term often introduces unreasonably large variance due to division by small probabilities appearing in the correction matrix (see Definition 1). To grant ourselves more flexibility in balancing bias and variance, we propose and analyze a family of $\gamma$-corrected losses, parameterized by a matrix valued function $\gamma:\mathcal{X}\to\mathbb{R}^k\times\mathbb{R}^k$:

$$\ell_\gamma(z;f(x),p(x))\triangleq\ell(z;f(x),p(x))+(y-p(x))^\top\gamma(x)f(x)$$

to mimic the bilinear structure of Bregman orthogonal losses (2). Note that we can always recover the vanilla distillation loss by taking $\gamma\equiv 0$. We denote the associated population and empirical risks by

$$L_D(f,p,\gamma)\triangleq\mathbb{E}[\ell_\gamma(Z;f(X),p(X))] \quad\text{and}\quad L_n(f,p,\gamma)\triangleq\mathbb{E}_n[\ell_\gamma(Z;f(X),p(X))].$$

Observe that at $p_0$ the correction term is mean-zero and hence $L_D(f,p_0,\gamma)$ is independent of $\gamma$

$$L_D(f,p_0)\triangleq\mathbb{E}[\ell(Z;f(X),p_0(X))]=L_D(f,p_0,\gamma) \quad\text{for all}\quad \gamma.$$

The $\gamma$-corrected loss has strong connections to the literature on Neyman orthogonality (Chernozhukov et al., 2018; Chernozhukov et al., 2016; Nekipelov et al., 2018; Chernozhukov et al., 2018; Foster & Syrgkanis, 2019). In particular, if the function $\gamma$ is set appropriately, then one can show that the $\gamma$-corrected loss function satisfies the condition of a Neyman orthogonal loss defined by Foster & Syrgkanis (2019). We begin our analysis by showing a general lemma for any estimator $\hat{f}$, which adapts the main theorem of Foster & Syrgkanis (2019) to account for approximate orthogonality; the proof can be found in App. F.

**Lemma 4** (Algorithm-agnostic analysis). *Consider any estimation algorithm that produces an estimate $\hat{f}$ with small plug-in excess risk, i.e.,*

$$L_D(\hat{f},\hat{p},\gamma)-L_D(f_0,\hat{p},\gamma)\leq\epsilon(\hat{f},\hat{p},\gamma).$$

*If the loss $L_D$ is $\sigma$-strongly convex with respect to $f$ and $\mathcal{F}$ is a convex set, then*

$$\frac{\sigma}{4}\|\hat{f}-f_0\|_{2,2}^2\leq\epsilon(\hat{f},\hat{p},\gamma)+\frac{1}{\sigma}\|(\gamma_{f_0,\hat{p}}-\gamma)^\top(\hat{p}-p_0)\|_{2,2}^2.$$

*If, in addition, $\sup_{z,\phi,\pi,i\in[d]}\|\nabla_{\phi_i\pi\pi}\ell(z;\phi,\pi)\|_{\mathrm{op}}\leq M$, then*

$$\|(\gamma_{f_0,\hat{p}}-\gamma)^\top(\hat{p}-p_0)\|_{2,2}^2\leq 2\big(\|(q_{f_0,\hat{p}}-\gamma)^\top(\hat{p}-p_0)\|_{2,2}^2+M^2 k\|\hat{p}-p_0\|_{2,4}^4\big).$$

**Connection to Neyman orthogonality** Remarkably, if we set $\gamma=q_{f_0,\hat{p}}$, then the $\gamma$-corrected loss is Neyman orthogonal (Foster & Syrgkanis, 2019), and the student MSE bound depends only on the *squared* MSE of the teacher. Moreover, $q_{f_0,\hat{p}}$ is an observable quantity for any Bregman divergence loss (Definition 1) as $q_{f_0,\hat{p}}$ is independent of $f_0$. However, we note that this setting of the $\gamma$ can lead to larger variance, i.e., the achievable excess risk can be much larger than the excess risk without the correction. For instance, in the case of the SEL loss $q_{f_0,\hat{p}}(x)=\frac{1}{\hat{p}(x)}$, which can be excessively large when $\hat{p}$ is close to 0, leading to a large increase in the variance of our loss. Thus, in a departure from the standard approach in semiparametric inference, we will be choosing $\gamma$ in practice to balance bias and variance.

**Example instantiation of student's estimation algorithm**  If we use *plug-in empirical risk minimization*, i.e., $\hat{f} = \text{argmin}_{f \in \mathcal{F}} L_n(f, \hat{p}, \gamma)$, to estimate $f_0$ with $\hat{p}$ estimated on an independent sample, then the results of Maurer & Pontil (2009) directly imply that as long as the loss function $\ell(z; \phi, \pi)$ is uniformly bounded in $[-H, H]$, then, with probability at least $1 - \delta$,

$$\epsilon(\hat{f}, \hat{p}, \gamma) = O\left(\sqrt{\frac{\sup_{f \in \mathcal{F}} \text{Var}(\ell_\gamma(Z; f(X), \hat{p}(X))) \log(\tau(n)/\delta)}{n}} + \frac{H \log(\tau(n)/\delta)}{n}\right)$$

where $\tau(n) = \mathcal{N}_\infty(1/n, \mathcal{F}, 2n)$ and $\mathcal{N}_\infty(\epsilon, \mathcal{F}, m)$ is the $\ell_\infty$ empirical covering number of function class $\mathcal{F}$ in the worst-case over all realizations of $m$ data points and at approximation level $\epsilon$. This result has two drawbacks: it is a slow rate result that scales as $1/\sqrt{n}$ for parametric or bounded Vapnik–Chervonenkis (VC)-dimension classes, and it requires the student to be fit on a completely separate dataset from the teacher's. In the next theorem, we address both of these drawbacks: i) we invoke localized Rademacher complexity analysis to provide a fast rate result which would be of the order of $1/n$ for VC or parametric function classes, and ii) we use a more sophisticated data-partitioning technique called cross-fitting, which allows the student to be trained using all of the available teacher data.

## 4.2 COMBATING TEACHER OVERFITTING WITH CROSS-FITTING

We now describe a more sophisticated version of data partitioning to make use of all data points in our student estimation, while at the same time not suffering from the sample complexity of the teacher's function space. This approach is referred to as *cross-fitting* (CF) in the semiparametric inference literature (see, e.g., Chernozhukov et al. (2018)):

1. Partition the dataset into $B$ equally sized folds $P_1, ..., P_B$.
2. For each fold $t \in [B]$ estimate $\hat{p}^{(t)}$ and $\hat{\gamma}^{(t)}$ using all the out-of-fold data points.
3. Estimate $\hat{f}$ by minimizing the empirical loss:
$$\hat{f} = \text{argmin}_{f \in \mathcal{F}} \frac{1}{n} \sum_{t=1}^{B} \sum_{i \in P_t} \ell_{\hat{\gamma}^{(t)}}(Z_i; f(X_i), \hat{p}^{(t)}(X_i)). \qquad \text{(Enhanced KD)}$$

In other words, the nuisance estimates $(\hat{\gamma}^{(t)}, \hat{p}^{(t)})$ that are evaluated on the data points in fold $t$ when fitting the student in step 3, are estimated only using data points outside of $P_t$.

**Theorem 5** (Enhanced KD analysis). *Suppose $f_0$ belongs to a convex set $\mathcal{F}$. Let $\delta_{n/B, \zeta/B} = \delta_{n/B} + c_0 \sqrt{\frac{B \log(c_1 B/\zeta)}{n}}$ for universal constants $c_0, c_1$ and $\delta_{n/B}$ an upper bound on the critical radius of the class*
$$\mathcal{G}(\hat{p}^{(t)}, \hat{\gamma}^{(t)}) = \{z \to r\left(\ell_{\hat{\gamma}^{(t)}}(z; f(x), \hat{p}^{(t)}(x)) - \ell_{\hat{\gamma}^{(t)}}(z; f_0(x), \hat{p}^{(t)}(x))\right) : f \in \mathcal{F}, r \in [0, 1]\}$$
*for each $t \in [B]$. Let $\mu(z) = \sup_{f \in \mathcal{F}, t \in [B]} \|\nabla_\phi \ell_{\hat{\gamma}^{(t)}}(z; f(X), \hat{p}^{(t)}(x))\|_2$, and assume that, with probability 1 for each $t \in [B]$, the loss $\ell_{\hat{\gamma}^{(t)}}(z; \phi, \hat{p}^{(t)}(x))$ is $\sigma$-strongly convex in $\phi$ for each $z$ and each $g \in \mathcal{G}(\hat{p}^{(t)}, \hat{\gamma}^{(t)})$ is uniformly bounded in $[-H, H]$. Moreover, suppose that the function class $\mathcal{F}$ satisfies the $\ell_2/\ell_4$ ratio condition: $\sup_{f \in \mathcal{F}} \frac{\|f - f_0\|_{2,4}}{\|f - f_0\|_{2,2}} \leq C$. If $\hat{f}$ is the output of Enhanced KD, then, with probability at least $1 - \zeta$,*

$$\frac{\sigma}{8} \|\hat{f} - f_0\|_{2,2}^2 = \frac{1}{\sigma} O\left(\delta_{n/B, \zeta/B}^2 C^2 H^2 \left(\|\mu\|_4^2 + \frac{1}{B} \sum_{t=1}^{B} \sqrt{\mathbb{E}\left[\|(Y - \hat{p}^{(t)}(X))^\top \hat{\gamma}^{(t)}(X)\|_2^4\right]}\right)\right)$$
$$+ \frac{1}{\sigma} O\left(\frac{1}{B} \sum_{t=1}^{B} \|(\gamma_{f_0, \hat{p}^{(t)}} - \hat{\gamma}^{(t)})^\top (\hat{p}^{(t)} - p_0)\|_{2,2}^2\right).$$

The proof is found in App. G. Observe that, unlike Thm. 1, the function classes $\mathcal{G}(\hat{p}^{(t)}, \hat{\gamma}^{(t)})$ in the Thm. 5 do not vary the teacher's model over $\mathcal{P}$ but rather evaluate $p$ at the specific out-of-fold estimates $\hat{p}^{(t)}$ and only vary $f \in \mathcal{F}$. Since in practice the teacher's model can be quite complex, removing this dependence on the sample complexity of the teacher's function space can bring immense improvement with the critical radius of $\mathcal{G}(\hat{p}^{(t)}, \hat{\gamma}^{(t)})$ significantly smaller than that of $\mathcal{G}$ from Thm. 1.

For instance, suppose that the loss function $\ell_{\hat{\gamma}^{(t)}}(z; f, \hat{p}^{(t)})$ is $L$-Lipschitz with respect to $f$ and that $\mathcal{F}$ is a VC-subgraph class with VC dimension $d_\mathcal{F}$. Then the critical radius of the function class $\mathcal{G}(\hat{p}^{(t)}, \hat{\gamma}^{(t)})$ is of order $\sqrt{d_\mathcal{F} \log(n)/n}$ for any choice of $(\hat{p}^{(t)}, \hat{\gamma}^{(t)})$ (see, e.g., Foster & Syrgkanis, 2019, Sec. 4.2).[2] However, under the same conditions, the critical radius of the teacher-student function class $\mathcal{G}$ in

---

[2]In fact, under the Lipschitz condition alone and using contraction lemma arguments as in Foster & Syrgkanis (2019, Lem. 11), one can derive a version of Thm. 5 in which the upper bound depends only on the critical radius of the function class $\{r(f - f_0) : f \in \mathcal{F}, r \in [0, 1]\}$, which solely depends on the function space of the student.

Thm. 1 will still depend on the teacher's function space. If $\mathcal{P}$ is also a VC-subgraph class with VC dimension $d_{\mathcal{P}} \gg d_{\mathcal{F}}$, then the critical radius of $\mathcal{G}$ will be of the much larger order $\sqrt{d_{\mathcal{P}} \log(n)/n}$.

We can also see in the bound of Thm. 5 the interplay between bias and variance introduced by $\gamma$. In particular, the part of the bound that depends on $\hat{\gamma}^{(t)}$ can be further simplified as

$$\sqrt{\mathbb{E}[\delta_{n,\zeta}^4 C^4 \|(Y-\hat{p}(X))^\top \hat{\gamma}^{(t)}(X)\|_2^4 + \|(\gamma_{\hat{p},0}(X) - \hat{\gamma}^{(t)}(X))^\top (\hat{p}(X) - p_0(X))\|_2^4]}, \quad (3)$$

where the terms respectively encode the increase in variance and decrease in bias from employing loss correction. Notably, Thm. 5 implies that CF without $\gamma$-correction (i.e., $\hat{\gamma}^{(t)}(x) = 0$) is sufficient to reduce student error due to teacher overfitting but may still be susceptible to excessive student error due to teacher underfitting. These qualitative predictions accord with our experimental observations in Sec. 5 and Fig. 5.

### 4.3 BIASED STOCHASTIC GRADIENT DESCENT ANALYSIS

When the set of candidate prediction rules $f_\theta$ is parameterized by a vector $\theta \in \mathbb{R}^d$, we may alternatively fit $\theta$ via stochastic gradient descent (SGD) (Robbins & Monro, 1951; Bottou & Bousquet, 2008) on the $\gamma$-corrected objective $L_D(f_\theta, \hat{p}, \hat{\gamma})$. With a minibatch size of 1 and a starting point $\theta_0$, the parameter updates take the form

$$\theta_{t+1} = \theta_t - \eta_t \nabla_\theta f_\theta(X_t)^\top \nabla_\phi \ell_\gamma(W_t; f_\theta(X_t), p(X_t)) \quad \text{for} \quad t+1 \in [n]. \quad (4)$$

Ideally, these updates would converge to a minimizer of the ideal risk $\mathcal{L}(\theta; p_0) = L_D(f_\theta, p_0)$. Our next result shows that, if the teacher $\hat{p}$ is independent of $(W_t)_{t \in [n]}$, then the SGD updates (4) have excess ideal risk governed by a bias term $\zeta(\hat{\gamma})$ and a variance term $\sigma(\hat{\gamma})^2/n$. Here, $\sigma_0^2(\theta)$ represents the baseline stochastic gradient variance that would be incurred if SGD were run directly on the ideal risk $\mathcal{L}(\theta; p_0)$ rather than our surrogate risk. Our proof in App. H builds upon the biased SGD bounds of Ajalloeian & Stich (2020).

**Theorem 6** (Biased SGD analysis). *Suppose that the loss $\mathcal{L}(\theta; p_0)$ is $\lambda$-strongly smooth in $\theta$. Define the bias and root-variance parameters*

$$\zeta(\hat{\gamma}) \triangleq \sup_{\theta \in \mathbb{R}^d} \|\nabla_\theta f_\theta^\top (\gamma_{f_\theta, \hat{p}} - \hat{\gamma})^\top (\hat{p} - p_0)\|_{2,2}$$

$$\sigma(\hat{\gamma}) \triangleq \sup_{\theta \in \mathbb{R}^d} \sigma_0(\theta) + \sqrt{\mathbb{E}[\|\nabla_\theta f_\theta(X)^\top \gamma(X)^\top (Y - p_0(X))\|_2^2] + \|\nabla_\theta f_\theta^\top (\gamma_{f_\theta, \hat{p}} - \hat{\gamma})^\top (\hat{p} - p_0)\|_{2,2}^2}$$

*for $\sigma_0^2(\theta) \triangleq \sum_{i \in [d]} \text{Var}[\nabla_{\theta_i} \ell(W; f_\theta(X), p_0(X))]$ the unbiased SGD variance. If $F_0 = \mathcal{L}(\theta_0; p_0) - \min_{\theta \in \mathbb{R}^d} \mathcal{L}(\theta; p_0)$, then the iterates $\{\theta_t\}_{t=1}^n$ of the loss corrected SGD algorithm satisfy*

$$\min_{t \in [n]} \mathbb{E}[\|\nabla_\theta \mathcal{L}(\theta_t; p_0)\|_2^2] = O\left(\frac{\sigma(\hat{\gamma})\sqrt{\lambda F_0}}{\sqrt{n}} + \zeta^2(\hat{\gamma})\right).$$

*If, in addition, $\mathcal{L}(\theta; p_0)$ is $\mu$-strongly convex in $\theta$, then the iterates satisfy*

$$\mathbb{E}[\mathcal{L}(\theta_n; p_0) - \min_{\theta \in \mathbb{R}^d} \mathcal{L}(\theta; p_0)] = \frac{1}{\mu} O\left(\frac{\lambda}{\mu} \frac{\sigma(\hat{\gamma})^2}{n} + \zeta^2(\hat{\gamma})\right) + O\left(F_0 e^{-\frac{\mu}{2\lambda} n}\right).$$

Similar to Thm. 5, the bound in Thm. 6, portrays the interplay of bias and variance as $\hat{\gamma}$ ranges from 0 to $q_{f_\theta, \hat{p}}$ (recall that $q_{f_\theta, \hat{p}}$ is independent of $f_\theta$ for any Bregman loss). In particular, the part of the bound for strongly convex losses that depends on $\hat{\gamma}$ can be further simplified to:

$$\mathbb{E}\left[\left(\frac{\lambda \|\hat{\gamma}(X)\|_2^2 \|Y - p_0(X)\|_2^2}{\mu n} + \|(\gamma_{f_\theta, \hat{p}}(X) - \hat{\gamma}(X))^\top (\hat{p}(X) - p_0(X))\|_2^2\right) \|\nabla_\theta f_{\hat{\theta}}(X)\|_2^2\right] \quad (5)$$

This has a very intuitive form: the first term is the impact of $\hat{\gamma}(X)$ on the variance, which is also related to the square of the noise of $y$, divided by the standard error scaling. The second controls how $\hat{\gamma}$ improves the bias introduced by the error in the teacher's $\hat{p}$.

## 5 EXPERIMENTS

We complement our theoretical analysis with a pair of experiments demonstrating the practical benefits of cross-fitting and loss correction on six real-world classification tasks. Throughout, we use the SEL loss and report mean performance $\pm 1$ standard error across 5 independent runs. Code to replicate all experiments can be found at

```
https://github.com/microsoft/semiparametric-distillation,
```

and supplementary experimental details and results can be found in App. I.

**Selecting the loss correction matrix** $\hat{\gamma}$ Motivated by the analyses in Sec. 4, for each training point $(x,y)$, we will select our correction matrix $\hat{\gamma}(x)$ to balance bias and variance by minimizing a pointwise upper bound on the loss correction error (5) (ideally with a closed-form solution to avoid excessive computational overhead).[3]  To eliminate dependence on the unobserved $p_0$, we observe that the bias term $\|(\gamma_{f_\theta,\hat{p}}(x) - \hat{\gamma}(x))^\top (\hat{p}(x) - p_0(x))\|_2^2 = O(\|q_{f_\theta,\hat{p}}(x) - \hat{\gamma}(x)\|_{\text{op}}^2)$ up to additive terms independent of $\hat{\gamma}$. We introduce a tunable hyperparameter $\alpha > 0$ to trade off between this bias bound and the variance term in (5) and select $\hat{\gamma}(x) = \text{diag}(v(x))$ to minimize:

$$\mathbb{E}[\|\hat{\gamma}(x)(y - \hat{p}(x))\|_2^2 \,|\, x] + \alpha \|q_{f_\theta,\hat{p}}(x) - \hat{\gamma}(x)\|_{\text{op}}^2 = \mathbb{E}[\|v(x)(y - \hat{p}(x))\|_2^2 \,|\, x] + \alpha \|\tfrac{1}{\hat{p}(x)} - v(x)\|_2^2.$$

Since the conditional expectation involves the unknown quantity $p_0$, we estimate $\mathbb{E}[\|v(x)(y - \hat{p}(x))\|_2^2 \mid x]$ with its sample $\|v(x)(y - \hat{p}(x))\|_2^2$.[4]  This objective is quadratic in $v(x)$ and thus has a closed-form solution. Given $\hat{\gamma}(x)$, the student's loss-corrected objective is equivalent to a square loss with labels $\log(p(x)) + \hat{\gamma}(x)^\top (y - p(x))$.

**Tabular data.** We first validate our KD enhancements on five real-world tabular datasets—FICO (FIC), StumbleUpon (Eve; Liu et al., 2017), and Adult, Higgs, and MAGIC from Dheeru & Karra Taniskidou (2017)—with random forest (Breiman, 2001) students and teachers. In Fig. 1a, we examine the impact of varying student model capacity for a fixed high-capacity teacher with 500 trees on FICO. This setting lends itself to teacher overfitting, and we find that cross-fitting consistently improves upon vanilla KD by up to 4 AUC percentage points. In Fig. 1b we explore the impact of teacher underfitting by limiting the teacher's maximum tree depth on Adult. Here we observe consistent gains from loss correction with student performance exceeding even that of the teacher for smaller maximum tree depths. Analogous results for the remaining datasets can be found in App. I.1.

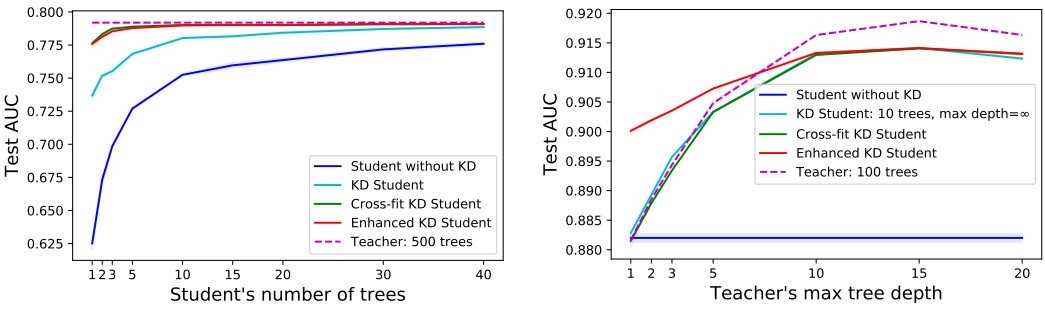

(a) FICO dataset, when teacher overfits      (b) Adult dataset, when teacher underfits

Figure 1: For random forest students and teachers, cross-fitting improves student performance when the teacher overfits, while loss correction improves student performance when the teacher underfits.

**Image data.** We next validate our KD enhancements on the image classification dataset CIFAR-10 (Krizhevsky & Hinton, 2009). We pair a residual network (ResNet-8) student with teacher networks of varying depths (ResNet-14/20/32/44/56) (He et al., 2016). It has been observed that larger and deeper teachers need not yield better students, as the teacher might overfit to the training set (Cho & Hariharan, 2019; Müller et al., 2019). To induce this overfitting, we turn off data augmentation (random horizontal flipping and cropping). We compare students trained with Vanilla KD and Enhanced KD with and without loss correction in Fig. 2. We find that cross-fitting consistently reduces the effect of teacher overfitting with largest impact realized for the deepest models. This effect is most evident in the cross-entropy test loss, where the Vanilla KD student incurs significantly larger loss than the cross-fitted student. For both accuracy and test loss, employing loss correction on top of cross-fitting provides an additional small performance boost.

**Effect of the loss correction hyperparameter** $\alpha$. Our hyperparameter $\alpha$ controls the tradeoff between bias and variance in loss correction. When $\alpha$ is very small, the objective is close to the vanilla KD objective. When $\alpha$ is large, the objective is closer to the Neyman-orthogonal loss. In Figure 3, we show the effect of varying $\alpha$, with ResNet-8 as the student and ResNet-20 as the teacher, on the CIFAR-10 dataset. Large values of $\alpha$ lead to high variance and thus lower test accuracy. Intermediate values of $\alpha$ improves on both the Vanilla KD objective, which corresponds to $\alpha = 0$ and on the orthogonal objective

---

[3]Balancing the bias and variance terms (3) of Thm. 5 yields a similar objective.

[4]An alternative estimate that performs slightly worse is $\|v(x)\hat{p}(x)(1 - \hat{p}(x))\|_2^2$.

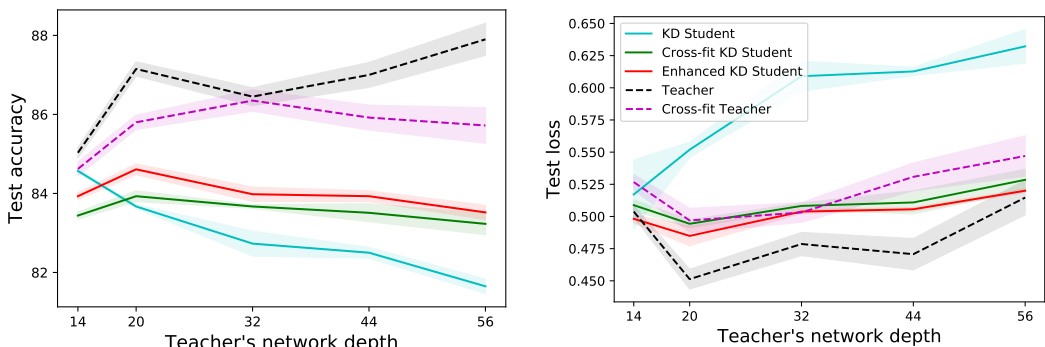

Figure 2: On CIFAR-10 with ResNet students and teachers, cross-fitting reduces the effect of teacher overfitting, and loss correction yields an additional small performance boost. Here, the test loss is cross-entropy.

($\alpha = \infty$). The test accuracy drops sharply beyond some threshold of $\alpha$ as the variance becomes too high (due to the terms $q_{\hat{p}}(x) = \mathrm{diag}\left(\frac{1}{\hat{p}_1(x)}, ..., \frac{1}{\hat{p}_K(x)}\right)$), causing training to become unstable.

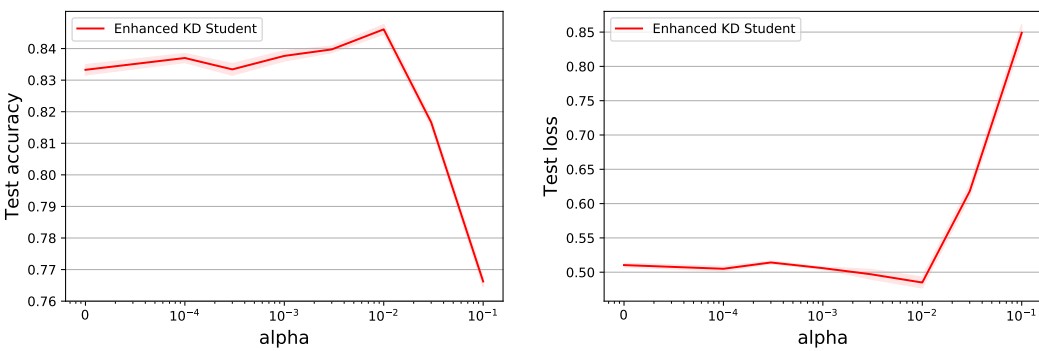

Figure 3: On CIFAR-10 with ResNet students and teachers, large values of the loss correction hyperparameter $\alpha$ (corresponding to the orthogonal loss correction) lead to large variance and training instability, while intermediate values improve upon cross-fit KD without loss correction ($\alpha = 0$). Here, the test loss is cross-entropy.

## 6 CONCLUSION

We developed a new analysis of knowledge distillation under the lens of semiparametric inference. By framing the KD process as learning with plug-in estimation in the presence of nuisance, we obtained new generalization bounds for distillation and new lower bounds highlighting the susceptibility of KD to teacher overfitting and underfitting. To address these failure modes, we introduced two complementary KD enhancements—cross-fitting and loss correction—which improve student performance both in theory and in practice. Past work has shown that augmenting the student training set with synthetic data from a generative model (e.g., a generative adversarial network (Liu et al., 2018) or MUNGE (Bucila et al., 2006)) often leads to improved student performance. A natural next step is to prove an analogue of Thm. 5 for synthetic augmentation to understand when this strategy successfully mitigates the impact of teacher overfitting. In addition, two tantalizing open questions are, first, whether other techniques from semiparametric inference, such as targeted maximum likelihood (Van Der Laan & Rubin, 2006), can be used to improve KD performance and, second, whether a semiparametric perspective can explain the surprising success of self-distillation (Furlanello et al., 2018) and noisy student training (Xie et al., 2020) through which students routinely outperform their teachers.

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

## A  EXTENDED LITERATURE REVIEW

We point the interested reader to Gou et al. (2020) for a sweeping survey of the many developments in knowledge distillation over the past half decade. In addition to the references discussing theoretical aspects of knowledge distillation provided in Sec. 1, we highlight here a number of empirical investigations of why distillation works. Cho & Hariharan (2019) show that larger teacher models do not necessarily improve the performance of student models as parsimonious student models are not able to mimic the teacher model. They suggest early stopping in training large teacher neural networks as means of regularizing. Cheng et al. (2020) demonstrate that when applied to image data, distillation allows the student neural net to learn multiple visual concepts simultaneously, while, when learning from raw data, neural networks learn concepts sequentially.

Knowledge distillation has also been used for adversarial attacks (Papernot et al., 2016b; Ross & Doshi-Velez, 2017; Gil et al., 2019; Goldblum et al., 2020), data security (Papernot et al., 2016a; Lopes et al., 2017; Wang et al., 2019), image processing (Li & Hoiem, 2017; Wang et al., 2017; Chen et al., 2018; Li et al., 2017), natural language processing (Nakashole & Flauger, 2017; Mou et al., 2016; Hu et al., 2018; Freitag et al., 2017), and speech processing (Chebotar & Waters, 2016; Lu et al., 2017; Watanabe et al., 2017; Oord et al., 2018; Shen et al., 2018).

## B  GLOSSARY

Table 1: Glossary of notation

| Notation | Definition |
|---|---|
| $\ell(Z;f(X),p_0(X))$ | Loss function on a random data point |
| Population risk $L_D(f,p)$ | $\mathbb{E}[\ell(Z;f(X),p(X))]$ |
| Empirical risk $L_n(f,p)$ | $\mathbb{E}_n[\ell(Z;f(X),p(X))]$ |
| Population optimal student model $f_0$ | $\operatorname{argmin}_{f \in \mathcal{F}} L_D(f,p_0)$ |
| Empirical optimal student model $\hat{f}$ | $\operatorname{argmin}_{f \in \mathcal{F}} L_n(f,\hat{p})$ |
| $\|f\|_{p,q}$ | $\big\| \|f(X)\|_p \big\|_{L^q} = \mathbb{E}\big[\|f(X)\|_p^q\big]^{1/q}$ |
| $\nabla_\phi$ | Partial derivative of $\ell(z;\phi,\pi)$ with respect to the second input |
| $\nabla_\pi$ | Partial derivative of $\ell(z;\phi,\pi)$ with respect to the third input |
| $\nabla_{\phi\pi}$ | $[\nabla_{\phi\pi}\ell(z;\phi,\pi)]_{i,j} = \frac{\partial^2}{\partial\phi_j\partial\pi_i}\ell(z;\phi,\pi)$ |
| $q_{f,p}(x)$ | $\mathbb{E}[\nabla_{\phi\pi}\ell(Z;f(X),p(X))\,|\,X=x]$ |
| $\gamma_{f,p}(x)$ | $\mathbb{E}_{U\sim\mathrm{Unif}([0,1])}[q_{f,Up+(1-U)p_0}(x)]$ |
| $\mathcal{R}(\delta;\mathcal{F})$ | Localized Rademacher complexity of function class $\mathcal{F}$ |
| $\delta_n$ | Critical radius |
| $\gamma$-corrected loss $\ell_\gamma(z;f(x),p(x))$ | $\ell(z;f(x),p(x))+(y-p(x))^\top\gamma(x)f(x)$ |
| Population $\gamma$-risk $L_D(f,p,\gamma)$ | $\mathbb{E}[\ell_\gamma(Z;f(X),p(X))]$ |
| Empirical $\gamma$-risk $L_n(f,p,\gamma)$ | $\mathbb{E}_n[\ell_\gamma(Z;f(X),p(X))]$ |

## C  PROOF OF THM. 1: VANILLA DISTILLATION ANALYSIS

Introduce the shorthand $\ell_{f,\hat{p}}(z) = \ell(z;f(x),\hat{p}(x))$. Since $\delta_n$ upper bounds the critical radius of the function class $\mathcal{G}$, the localized Rademacher analysis of Foster & Syrgkanis (2019, Lem. 11) implies[5]

$$\left| L_n(\hat{f},\hat{p}) - L_n(f_0,\hat{p}) - (L_D(\hat{f},\hat{p}) - L_D(f_0,\hat{p})) \right| \leq O\Big( H\delta_{n,\zeta}\|\ell_{\hat{f},\hat{p}} - \ell_{f_0,\hat{p}}\|_{2,2} + H\delta_{n,\zeta}^2 \Big)$$

with probability at least $1-\zeta$. Moreover, by Cauchy-Scwharz,

$$\|\ell_{\hat{f},\hat{p}} - \ell_{f_0,\hat{p}}\|_{2,2} \leq \|\mu\|_4 \|\hat{f} - f_0\|_{2,4}.$$

By the assumed $\ell_2/\ell_4$ ratio condition we therefore have

$$\epsilon(\hat{f},\hat{p},\gamma) \leq O\Big( \delta_{n,\zeta} CH\|\mu\|_4 \|\hat{f} - f_0\|_{2,2} + H\delta_{n,\zeta}^2 \Big).$$

Plugging this bound into Lemma 4 (which holds irrespective of whether data re-use, sample splitting, or cross-fitting is employed) and applying the arithmetic-geometric mean inequality yields

$$\frac{\sigma}{8}\|\hat{f} - f_0\|_{2,2}^2 \leq \frac{1}{\sigma} O\big( \delta_{n,\zeta}^2 C^2 H^2 \|\mu\|_4^2 + \|\gamma_{\hat{p},0}^\top(\hat{p} - p_0)\|_{2,2}^2 \big)$$

---

[5]We apply Foster & Syrgkanis (2019, Lem. 11) with $\mathcal{L}_g = g$ for $g \in \mathcal{G}$ with $g^* = 0$. Then we instantiate the concentration inequality for the choice $g = \ell_{\hat{f},\hat{p}} - \ell_{f_0,\hat{p}} \in \mathcal{G}$.

## D  PROOF OF PROP. 2: IMPACT OF TEACHER UNDERFITTING ON VANILLA DISTILLATION

Suppose that $p_0$ does not vary with $x$ and, for known $\epsilon > 0$, belongs to the set
$$\mathcal{P} = \{p : p_j(x) \in [\epsilon, 1], \forall x \in \mathcal{X}, j \in [k]\}.$$
As all quantities in this proof are independent of $x$, we will omit the dependence on $x$ whenever convenient.

Consider the constant teacher estimate $\hat{p} = \frac{\bar{y}}{1+\lambda} \vee \epsilon$ obtained via ridge regression with regularization strength $\lambda \leq 1$ and $\bar{y} \triangleq \frac{1}{n}\sum_{i=1}^n y_i$. A constant student prediction rule in
$$\mathcal{F} = \{f : f_j(x) \in [\log(\epsilon), 0], \forall x \in \mathcal{X}, j \in [k]\}$$
trained via Vanilla KD with SEL loss yields $\hat{f}(x) = \log(\hat{p})$.

Suppose that, unbeknownst to the teacher and student, the true $p_0$ satisfies the more stringent condition $p_{0,j} \geq 2\epsilon$ for all $j \in [k]$. Then the student satisfies
$$f_0 - \hat{f} = \log(p_0) - \log(\hat{p}) \geq \mathrm{diag}(\tfrac{1}{p_0})(p_0 - \hat{p}) = \gamma_{f_0, p_0}^\top (p_0 - \hat{p}) = \gamma_{f_0, p_0}^\top \left(\tfrac{\lambda p_0 - (\bar{y} - p_0)}{(1+\lambda)} + \min(0, \tfrac{\bar{y}}{1+\lambda} - \epsilon)\right)$$
$$= \gamma_{f_0, p_0}^\top \left(\tfrac{\lambda p_0 - (\bar{y} - p_0)}{(1+\lambda)} + \min(0, \tfrac{\bar{y} - p_0 + p_0 - (1+\lambda)\epsilon}{1+\lambda})\right) \geq \gamma_{f_0, p_0}^\top \tfrac{\lambda p_0 - |\bar{y} - p_0|}{(1+\lambda)}$$
by the concavity of the logarithm and the choice $\lambda \leq 1$. Since
$$P(|\bar{y}_j - p_{0,j}| \geq \theta p_{0,j}) \leq \tfrac{2\zeta}{k} \quad \text{for} \quad \theta \geq \sqrt{\tfrac{2(1-p_{0,j})}{np_{0,j}}\log(\tfrac{k}{\zeta})} + \tfrac{4}{3}\tfrac{1}{np_{0,j}}\log(\tfrac{k}{\zeta})$$
by Bernstein's inequality (Bernstein, 1946), we have
$$P(\|f_0 - \hat{f}\|_{2,2}^2 \geq \|\gamma_{f_0, p_0}^\top (p_0 - \hat{p})\|_{2,2}^2) \geq P(f_{0,j} - \hat{f}_j \geq 0, \forall j \in [k]) \geq P(\lambda p_{0,j} \geq |\bar{y}_j - p_{0,j}|, \forall j \in [k]) \geq 1 - 2\zeta$$
whenever
$$\sqrt{\tfrac{2}{n\epsilon}\log(\tfrac{k}{\zeta})} + \tfrac{4}{3}\tfrac{1}{n\epsilon}\log(\tfrac{k}{\zeta}) \leq \lambda \leq 1.$$

Moreover, since $\limsup\limits_{n\to\infty} \frac{\sqrt{n}|\bar{y}_j - p_{0,j}|}{\sqrt{2p_{0,j}(1-p_{0,j})\log\log(n)}} = 1$ with probability 1 by the law of the iterated logarithm, $\|\gamma_{f_0, p_0}^\top (p_0 - \hat{p})\|_{2,2}^2 = \Omega(\min(1, \lambda^2))$ with probability 1 whenever $1 \geq \lambda \geq \sqrt{\frac{2\log\log(n)}{n\epsilon}}$. The choice
$$\lambda = \min\left(1, \max\left(\tfrac{1}{n^{1/4}}, \sqrt{\tfrac{2\log\log(n)}{n\epsilon}}, \sqrt{\tfrac{2}{n\epsilon}\log(\tfrac{k}{\zeta})} + \tfrac{4}{3}\tfrac{1}{n\epsilon}\log(\tfrac{k}{\zeta})\right)\right) = \Theta(\tfrac{1}{n^{1/4}})$$
now yields the first two advertised claims.

The final claim follows directly from Thm. 5 with $B = O(1)$ as $\hat{\gamma}^{(t)} = \gamma_{f_0, \hat{p}}$ and the critical radius of $\mathcal{G}(\hat{p}^{(t)}, \hat{\gamma}^{(t)})$ satisfies $\delta_{n/B} = O(\sqrt{Bk/n})$ by Wainwright (2019, Ex. 13.8).

## E  PROOF OF PROP. 3: IMPACT OF TEACHER OVERFITTING ON VANILLA DISTILLATION

Suppose that $p_0$ has Lipschitz gradient and, for known $\epsilon > 0$, belongs to the set
$$\mathcal{P} = \{p : p_j(x) \in [\epsilon, 1], \forall x \in \mathcal{X}, j \in [k]\}.$$

Suppose moreover that $X \in \mathbb{R}^d$ has Lebesgue density bounded away from 0 and $\infty$ and that $\epsilon < \frac{1}{4}\frac{\mathbb{E}[p_{0,j}(X)(1-p_{0,j}(X))^2]}{\mathbb{E}[(1-p_{0,j}(X))/p_{0,j}(X)]}$ for each $j$. Consider the teacher estimates $\hat{p}_j(x) = \max(\epsilon, \tilde{p}_j(x))$ for $\tilde{p}$ the Nadaraya-Watson kernel smoothing estimator (Nadaraya, 1964; Watson, 1964)
$$\tilde{p}(x) \triangleq \begin{cases} y_i & \text{if} \quad x = x_i \\ \sum_{i=1}^n y_i K((x-x_i)/h)/\sum_{i=1}^n K((x-x_i)/h) & \text{otherwise} \end{cases}$$
with kernel $K(x) = \|x\|_2^{-a}\mathbb{I}[\|x\|_2 \leq 1]$, $a \in (0, d/2)$, and $h = n^{-1/(4+d)}$. By Belkin et al. (2019, Thm. 1), the teacher satisfies $\mathbb{E}[\|p_0 - \hat{p}\|_{2,2}^2] = O(n^{-4/(4+d)})$.

Now instantiate the notation of Thm. 1, and consider a student prediction rule trained to learn a constant prediction rule via Vanilla KD with the SEL loss and
$$\mathcal{F} = \{f : f(x) = f(x') \in [\log(\epsilon), 0]^k \text{ for all } x, x' \in \mathcal{X}\}. \tag{6}$$
Since $\tilde{p}$ exactly interpolates the observed labels (i.e., $\tilde{p}(x_i) = y_i$), the critical radius of the teacher-student function class $\mathcal{G}$ satisfies $\delta_n = \Omega(1)$. Moreover, since the student only has access to the teacher's training set probabilities, its estimate $\hat{f}(x) = \frac{1}{n}\sum_{i=1}^n \log(\max(y_i, \epsilon))$ is inconsistent for the optimal constant rule $f_0(x) = \mathbb{E}[\log(p_0(X))]$ as
$$f_{0,j}(x) - \mathbb{E}\hat{f}_j(x) = \mathbb{E}[\log(p_{0,j}(X)) - \log(\max(Y_j, \epsilon))] \geq \mathbb{E}[\tfrac{p_{0,j}(X) - \max(Y_j, \epsilon)}{p_{0,j}(X)} + \tfrac{(\max(Y_j, \epsilon) - p_{0,j}(X))^2}{2}]$$
$$= \mathbb{E}[\tfrac{p_{0,j}(X)(1-p_{0,j}(X))^2 + (1-p_{0,j}(X))(p_{0,j}(X)-\epsilon)^2}{2}] - \epsilon\mathbb{E}[\tfrac{1-p_{0,j}(X)}{p_{0,j}(X)}] \geq \mathbb{E}[\tfrac{p_{0,j}(X)(1-p_{0,j}(X))^2}{4}]$$

by Taylor's theorem with Lagrange remainder. This non-vanishing student error reflects the non-vanishing critical radius $\delta_n$ of the composite student-teacher function class $\mathcal{G}$ defined in Thm. 1; since the student function class $\mathcal{F}$ has low complexity, the complexity of $\mathcal{G}$ is driven by the highly flexible interpolating teacher.

Next, instantiate the notation of Thm. 5, and consider a student prediction rule $\hat{f}$ trained via Enhanced KD with SEL loss, $\hat{\gamma}^{(t)} = 0$, $B = O(1)$, and $\mathcal{F}$ (6). The critical radius of $\mathcal{G}(\hat{p}^{(t)}, \hat{\gamma}^{(t)})$ satisfies $\delta_{n/B} = O(\sqrt{Bk/n})$ by Wainwright (2019, Ex. 13.8). Moreover, each cross-fitted teacher satisfies $\mathbb{E}[\|p_0 - \hat{p}^{(t)}\|^2_{2,2}] = O(n^{-4/(4+d)})$ by Belkin et al. (2019, Thm. 1), so, by Chebyshev's and Jensen's inequalities, with probability at least $1 - \zeta/2$,

$$\|p_0 - \hat{p}^{(t)}\|_{2,2} \leq \mathbb{E}[\|p_0 - \hat{p}^{(t)}\|_{2,2}] + \sqrt{2B\mathrm{Var}(\|p_0 - \hat{p}^{(t)}\|_{2,2})/\zeta}$$

$$\leq (1 + \sqrt{2B/\zeta})\sqrt{\mathbb{E}[\|p_0 - \hat{p}^{(t)}\|^2_{2,2}]} = O(n^{-2/(4+d)}) \quad \text{for all} \quad t.$$

Therefore, Thm. 5 implies that

$$\|\hat{f} - f_0\|^2_{2,2} = O(\tfrac{1}{n} + \tfrac{1}{B}\textstyle\sum_{t=1}^{B}\|(\gamma_{f_0,\hat{p}^{(t)}})^\top(\hat{p}^{(t)} - p_0)\|^2_{2,2})$$

$$= O(\tfrac{1}{n} + \tfrac{1}{B}\textstyle\sum_{t=1}^{B}\|(\mathrm{diag}(\tfrac{1}{\hat{p}^{(t)}}))(\hat{p}^{(t)} - p_0)\|^2_{2,2})$$

$$= O(\tfrac{1}{n} + \tfrac{1}{B\epsilon^2}\textstyle\sum_{t=1}^{B}\|\hat{p}^{(t)} - p_0\|^2_{2,2}) = O(n^{-4/(4+d)})$$

with probability at least $1 - \zeta$.

## F  PROOF OF LEMMA 4: ALGORITHM-AGNOSTIC ANALYSIS

First we define for any functional $L(f)$ the Frechet derivative as:

$$D_f L(f)[\nu] = \frac{\partial}{\partial t} L(f + t\nu)\,|_{t=0}$$

When $L$ is an operator of the form: $\mathbb{E}[g(f(X))]$, then: $D_f L(f)[\nu] = \mathbb{E}[\nabla g(f(X))^\top \nu(X)]$.

By the $\sigma$-strong convexity of $L_D$,[6] we have that

$$L_D(\hat{f}, \hat{p}, \gamma) \geq L_D(f_0, \hat{p}, \gamma) + D_f L_D(f_0, \hat{p}, \gamma)[\hat{f} - f_0] + \frac{\sigma}{2}\|\hat{f} - f_0\|^2_{2,2}.$$

Furthermore, our excess risk assumption and the optimality of $f_0$ give us

$$\frac{\sigma}{2}\|\hat{f} - f_0\|^2_{2,2} \leq \underbrace{L_D(\hat{f}, \hat{p}, \gamma) - L_D(f_0, \hat{p}, \gamma)}_{\text{excess risk of } \hat{f}} - D_f L_D(f_0, \hat{p}, \gamma)[\hat{f} - f_0]$$

$$\overset{(a)}{\leq} \epsilon(\hat{f}, \hat{p}, \gamma) - \underbrace{D_f L_D(f_0, p_0, \gamma)[\hat{f} - f_0]}_{\geq 0 \text{ by optimality of } f_0} + D_f(L_D(f_0, p_0, \gamma) - L_D(f_0, \hat{p}, \gamma))[\hat{f} - f_0].$$

By Taylor's theorem with integral remainder,

$$\mathbb{E}[\langle \nabla_\phi \ell(W; f_0(x), p_0(x)) - \nabla_\phi \ell(W; f_0(x), \hat{p}(x)), \hat{f}(x) - f_0(x)\rangle \mid X = x] \tag{7}$$

$$= (p_0(x) - \hat{p}(x))^\top \gamma_{f_0,\hat{p}}(x)(\hat{f}(x) - f_0(x))$$

whenever $\nabla_{\phi\pi}\ell$ is well-defined. We can now invoke the expansion (7) and Cauchy-Schwarz to obtain the bound

$$D_f(L_D(f_0, p_0, \gamma) - L_D(f_0, \hat{p}, \gamma))[\hat{f} - f_0]$$

$$= \mathbb{E}[\langle \nabla_\phi \ell(W; f_0(X), p_0(X)) - \nabla_\phi \ell(W; f_0(X), p(X)), \hat{f}(X) - f_0(X)\rangle]$$

$$\quad - \mathbb{E}[(p_0(X) - \hat{p}(X))^\top \gamma(X)(\hat{f}(X) - f_0(X))]$$

$$= \mathbb{E}[(p_0(X) - \hat{p}(X))^\top (\gamma_{f_0,\hat{p}}(X) - \gamma(X))(\hat{f}(X) - f_0(X))]$$

$$\leq \mathbb{E}[\|(p_0(X) - \hat{p}(X))^\top (\gamma_{f_0,\hat{p}}(X) - \gamma(X))\|_2 \|\hat{f}(X) - f_0(X)\|_2]$$

$$\leq \|(p_0 - \hat{p})^\top (\gamma_{f_0,\hat{p}} - \gamma)\|_{2,2} \|\hat{f} - f_0\|_{2,2}$$

Thus combining all the above inequalities:

$$\frac{\sigma}{2}\|\hat{f} - f_0\|^2_{2,2} \leq \epsilon(\hat{f}, \hat{p}, \gamma) + \|(\hat{p} - p_0)^\top (\gamma_{f_0,\hat{p}} - \gamma)\|_{2,2} \|\hat{f} - f_0\|_{2,2}$$

---

[6]Notably this strong convexity assumption can be relaxed to $\mathbb{E}\left[\nabla_\phi \ell(W; f_0(X), p_0(X))(\hat{f}(X) - f_0(X))\right] \geq 0$.

By an AM-GM inequality, for all $a, b \geq 0$: $a \cdot b \leq \frac{1}{2}(\frac{2}{\sigma} a^2 + \frac{\sigma}{2} b^2)$. Applying this to the product of norms on the RHS and re-arranging yields

$$\frac{\sigma}{4} \|\hat{f} - f_0\|_{2,2}^2 \leq \epsilon(\hat{f}, \hat{p}, \gamma) + \frac{1}{\sigma} \|(\hat{p} - p_0)^\top (\gamma_{f_0, \hat{p}} - \gamma)\|_{2,2}^2.$$

To get the final inequality, observe that:

$$\|(\hat{p} - p_0)^\top (\gamma_{f_0, \hat{p}} - \gamma)\|_{2,2}^2 \leq 2\|(\hat{p} - p_0)^\top (q_{f_0, \hat{p}} - \gamma)\|_{2,2}^2 + 2\|(\hat{p} - p_0)^\top (\gamma_{f_0, \hat{p}} - q_{f_0, \hat{p}})\|_{2,2}^2$$

Moreover, by the boundedness of the third derivative, we have:

$$\|(\hat{p} - p_0)^\top (\gamma_{f_0, \hat{p}} - q_{f_0, \hat{p}})\|_{2,2}^2 \leq \mathbb{E}[\|\hat{p}(X) - p_0(X)\|_2^2 \|\gamma_{f_0, \hat{p}}(X) - q_{f_0, \hat{p}}(X)\|_2^2]$$

$$\leq \mathbb{E}[\|\hat{p}(X) - p_0(X)\|_2^2 M^2 k \|\hat{p}(X) - p_0(X)\|_2^2]$$

$$\leq M^2 k \|\hat{p} - p_0\|_{2,4}^4$$

Combining all the above yields the final bound.

## G   PROOF OF THM. 5: CROSS-FITTED ERM ANALYSIS

Let $L_{n,t}$ denote the empirical loss over the samples in the $t$-th fold and $\hat{p}^{(t)}, \hat{\gamma}^{(t)}$ the nuisance functions used on the samples in the $k$-th fold. For any $t \in [K]$ and conditional on $\hat{p}^{(t)}, \hat{\gamma}^{(t)}$, suppose that $\delta_n$ upper bounds the critical radius of the function class $\mathcal{G}(\hat{p}^{(t)}, \hat{\gamma}^{(t)})$, then by Lemma 11 of Foster & Syrgkanis (2019),[7] if we denote with $\ell_{t,f}(z) = \ell_{\hat{\gamma}^{(t)}}(z; f(x), \hat{p}^{(t)}(x))$, w.p. $1 - \zeta$:

$$\left| L_{n,t}(\hat{f}, \hat{p}^{(t)}, \hat{\gamma}^{(t)}) - L_{n,t}(f_0, \hat{p}^{(t)}, \hat{\gamma}^{(t)}) - (L_D(\hat{f}, \hat{p}^{(t)}, \hat{\gamma}^{(t)}) - L_D(f_0, \hat{p}^{(t)}, \hat{\gamma}^{(t)})) \right| \leq O\left( H\delta_{n/B, \zeta} \|\ell_{t,\hat{f}} - \ell_{t,f_0}\|_{2,2} + H\delta_{n/B, \zeta}^2 \right)$$

Moreover, we have that by the definition of cross-fitted ERM:

$$\frac{1}{B} \sum_{t=1}^{B} L_{n,t}(\hat{f}, \hat{p}^{(t)}, \hat{\gamma}^{(t)}) - L_{n,t}(f_0, \hat{p}^{(t)}, \hat{\gamma}^{(t)}) \leq 0$$

Thus we have that w.p. $1 - \zeta B$:

$$\frac{1}{B} \sum_{t=1}^{B} L_D(\hat{f}, \hat{p}^{(t)}, \hat{\gamma}^{(t)}) - L_D(f_0, \hat{p}^{(t)}, \hat{\gamma}^{(t)}) \leq O\left( H\delta_{n/B, \zeta} \frac{1}{B} \sum_{t=1}^{B} \|\ell_{t,f} - \ell_{t,f_0}\|_{2,2} + H\delta_{n/B, \zeta}^2 \right)$$

Moreover, if we let $\mu(z) = \sup_{\phi, t} \|\nabla_\phi \ell(z; \phi, \hat{p}^{(t)}(x))\|_2$, then we have by Cauchy-Schwarz inequality:

$$\|\ell_{t,f} - \ell_{t,f_0}\|_{2,2} \leq \|\mu\|_4 \|f - f_0\|_{2,4} + \sqrt{\mathbb{E}\left[ \left( (Y - \hat{p}^{(t)}(X))^\top \hat{\gamma}^{(t)}(X)(f(X) - f_0(X)) \right)^2 \right]}$$

$$\leq \|\mu\|_4 \|f - f_0\|_{2,4} + \mathbb{E}\left[ \left\| (Y - \hat{p}^{(t)}(X))^\top \hat{\gamma}^{(t)}(X) \right\|_2^2 \|f(X) - f_0(X)\|^2 \right]$$

$$\leq \left( \|\mu\|_4 + \mathbb{E}\left[ \left\| (Y - \hat{p}^{(t)}(X))^\top \hat{\gamma}^{(t)}(X) \right\|_2^4 \right]^{1/4} \right) \|f - f_0\|_{2,4}$$

If we further assume that the function class $\mathcal{F}$ satisfies an $\ell_2/\ell_4$ condition that:

$$\sup_{f \in \mathcal{F}} \frac{\|f - f_0\|_{2,4}}{\|f - f_0\|_{2,2}} \leq C$$

then w.p. $1 - \zeta$:

$$\frac{1}{B} \sum_{t=1}^{B} \epsilon(\hat{f}, \hat{p}^{(t)}, \hat{\gamma}^{(t)}) \leq O\left( H\delta_{n/B, \zeta/B} \frac{1}{B} \sum_{t=1}^{B} C\left( \|\mu_p\|_4 + \mathbb{E}\left[ \left\| (Y - \hat{p}(X))^\top \gamma(X) \right\|_2^4 \right]^{1/4} \right) \|f - f_0\|_{2,2} + H\delta_{n/B, \zeta/B}^2 \right).$$

Applying Lemma 4 for any $\hat{p}^{(t)}, \hat{\gamma}^{(t)}$ and averaging the final inequality we get:

$$\frac{\sigma}{4} \|\hat{f} - f_0\|_{2,2}^2 \leq \frac{1}{B} \sum_{t=1}^{B} \left( \epsilon(\hat{f}, \hat{p}^{(t)}, \hat{\gamma}^{(t)}) + \frac{1}{\sigma} \|(\gamma_{f_0, \hat{p}^{(t)}} - \hat{\gamma}^{(t)})^\top (\hat{p}^{(t)} - p_0)\|_{2,2}^2 \right).$$

---

[7] We apply the lemma with $\mathcal{L}_g = g$ and $g \in \mathcal{G}(\hat{p}^{(t)}, \hat{\gamma}^{(t)})$ and $g^* = 0$. Then we instantiate the concentration inequality with $g = \ell_{t,\hat{f}} - \ell_{t,f_0} \in \mathcal{G}(\hat{p}^{(t)}, \hat{\gamma}^{(t)})$.

Plugging in the bound above to Lemma 4 and applying the AM-GM inequality and Jensen's inequality, yields:

$$\frac{\sigma}{8}\|\hat{f}-f_0\|_{2,2}^2 \leq \frac{1}{\sigma}O\left(\delta_{n/B,\zeta/B}^2 C^2 H^2\left(\|\mu\|_4^2 + \frac{1}{B}\sum_{t=1}^{B}\sqrt{\mathbb{E}\left[\left\|(Y-\hat{p}^{(t)}(X))^\top\hat{\gamma}^{(t)}(X)\right\|_2^4\right]}\right)\right)$$
$$+\frac{1}{\sigma}O\left(\frac{1}{B}\sum_{t=1}^{B}\|(\gamma_{f_0,\hat{p}^{(t)}}-\hat{\gamma}^{(t)})^\top(\hat{p}^{(t)}-p_0)\|_{2,2}^2\right).$$

## H  PROOF OF THM. 6: BIASED SGD ANALYSIS

Below, for any integer $s$, we define the operator norm of any vector $v\in\mathbb{R}^s$ and any tensor $T$ operating on $\mathbb{R}^s$ as
$$\|v\|_{\mathrm{op}}\triangleq\|v\|_2 \quad\text{and}\quad \|T\|_{\mathrm{op}}\triangleq\sup_{v:\|v\|_2=1}\|T[v]\|_{\mathrm{op}}.$$

Recall the definition
$$\nabla(W;\theta,p,\gamma) = \nabla_\theta f_\theta(X)^\top\nabla_\phi\ell_\gamma(W;f_\theta(X),p(X))$$
$$= \nabla_\theta f_\theta(X)^\top(\nabla_\phi\ell(W;f_\theta(X),p(X))+\gamma(X)^\top(Y-p(X))).$$

Observe that since $\mathbb{E}[Y\,|\,X=x]=p_0(x)$, we can write for any $\gamma$:
$$\mathcal{L}(\theta;p_0)=\mathbb{E}[\ell(W;f_\theta(X),p_0(X))+(Y-p_0(X))^\top\gamma(X)f_\theta(X)]=\mathbb{E}[\ell_\gamma(W;f_\theta(X),p_0(X))]$$

Thus we also have that:
$$\forall\theta,\gamma:\nabla_\theta\mathcal{L}(\theta;p_0)=\mathbb{E}[\nabla(W;\theta,p_0,\gamma)]$$

Given this observation, we can decompose the gradient that is used in our SGD algorithm into a bias and variance component, when viewed from the perspective of a biased SGD algorithm for the population oracle loss:
$$\nabla(W;\theta,p,\gamma)=\nabla_\theta\mathcal{L}(\theta;p_0)$$
$$+\underbrace{\mathbb{E}[\nabla(W;\theta,p,\gamma)]-\mathbb{E}[\nabla(W;\theta,p_0,\gamma)]}_{\mathbf{b}(\theta,p,\gamma)}+\underbrace{\nabla(W;\theta,p,\gamma)-\mathbb{E}[\nabla(W;\theta,p,\gamma)]}_{\mathbf{n}(W;\theta,p,\gamma)}$$

The following two lemmas bound the gradient bias and noise terms.

**Lemma 7** (Gradient bias). *If $\sup_{x,\phi,\pi}\|\mathbb{E}[\nabla_{\pi\pi\phi}\ell(W;\phi,\pi)\,|\,X=x]\|_{\mathrm{op}}\leq M$, then for any parameter vector $\theta$ and functions $p$ and $\gamma$, we have:*
$$\mathbf{b}(\theta,p,\gamma)=\mathbb{E}[\nabla_\theta f_\theta(X)^\top(\gamma_{f_\theta,p}(X)-\gamma(X))^\top(p(X)-p_0(X))],$$
$$\|\mathbf{b}(\theta,p,\gamma)\|_2\leq\left\|\nabla_\theta f_\theta^\top(\gamma_{f_\theta,p}-\gamma)^\top(p-p_0)\right\|_{2,2}, \quad\text{and}$$
$$\|\mathbf{b}(\theta,p,\gamma)\|_2\leq\left\|\nabla_\theta f_\theta^\top(q_{f_\theta,p}-\gamma)^\top(p-p_0)\right\|_{2,2}+\frac{M}{2}\|\nabla_\theta f_\theta\|_{F,2}\|p-p_0\|_{2,4}^2.$$

**Proof**  By Taylor's theorem with integral remainder and Lagrange remainder respectively the SGD bias for each parameter $i$ takes the form
$$\mathbf{b}_i(\theta,p,\gamma)=\mathbb{E}[\nabla_i(W;\theta,p,\gamma)]-\mathbb{E}[\nabla_i(W;\theta,p_0,\gamma)]$$
$$=\mathbb{E}[\langle\nabla_\phi\ell_\gamma(W;f_\theta(X),p(X))-\nabla_\phi\ell_\gamma(W;f_\theta(X),p_0(X)),\nabla_{\theta_i}f_\theta(X)\rangle]$$
$$=\mathbb{E}[(p(X)-p_0(X))^\top(\gamma_{f_\theta,p}(X)-\gamma(X))\nabla_{\theta_i}f_\theta(X)]$$
$$=\mathbb{E}[(p(X)-p_0(X))^\top(q_{f_\theta,p}(X)-\gamma(X))\nabla_{\theta_i}f_\theta(X)]$$
$$+\frac{1}{2}\mathbb{E}[\nabla_{\pi\pi\phi}\ell(W;f_\theta(X),\bar{p}(X))[\nabla_{\theta_i}f_\theta(X),p(X)-p_0(X),p(X)-p_0(X)]].$$

Furthermore, our operator norm assumption and Cauchy-Schwarz imply
$$|\mathbf{b}_i(\theta,p,\gamma)|\leq|\mathbb{E}[(p(X)-p_0(X))^\top(q_{f_\theta,p}(X)-\gamma(X))\nabla_{\theta_i}f_\theta(X)]|+\frac{M}{2}\mathbb{E}\left[\|\nabla_{\theta_i}f_\theta(X)\|_2\|p(X)-p_0(X)\|_2^2\right]$$
$$\leq|\mathbb{E}[(p(X)-p_0(X))^\top(q_{f_\theta,p}(X)-\gamma(X))\nabla_{\theta_i}f_\theta(X)]|+\frac{M}{2}\|\nabla_{\theta_i}f_\theta\|_{2,2}\|p-p_0\|_{2,4}^2.$$

Thus, by the triangle inequality and Jensen's inequality we find that
$$\|\mathbf{b}(\theta,p,\gamma)\|_2\leq\left\|\nabla_\theta f_\theta^\top(\gamma_{f_\theta,p}-\gamma)^\top(p-p_0)\right\|_{2,2} \quad\text{and}$$
$$\|\mathbf{b}(\theta,p,\gamma)\|_2\leq\left\|\nabla_\theta f_\theta^\top(q_{f_\theta,p}-\gamma)^\top(p-p_0)\right\|_{2,2}+\frac{M}{2}\|\nabla_\theta f_\theta\|_{F,2}\|p-p_0\|_{2,4}^2.$$
$\square$

**Lemma 8** (Gradient Variance). *Define For any parameter $\theta$ and functions $p$ and $\gamma$,*
$$\sqrt{\mathbb{E}[\|\mathbf{n}(W;\theta,p,\gamma)\|_2^2]}\leq\sigma_0(\theta)+\sqrt{\mathbb{E}[\|\nabla_\theta f_\theta(X)^\top\gamma(X)^\top(Y-p_0(X))\|_2^2]}+\left\|\nabla_\theta f_\theta^\top(\gamma_{f_\theta,p}-\gamma)^\top(p-p_0)\right\|_{2,2}^2.$$

**Proof** For each $i \in [d]$, define the shorthand

$$\Delta_i = \nabla_{\theta_i} \ell(W; f_\theta(X), p(X)) + \nabla_{\theta_i} f_\theta(X)^\top \gamma(X)^\top (Y - p(X)) - \nabla_{\theta_i} \ell(W; f_\theta(X), p_0(X)) \quad \text{and}$$

$$\begin{aligned}
Z_i &= \mathbb{E}[\Delta_i \mid X] \\
&= \nabla_{\theta_i} f_\theta(X)^\top (\gamma(X) - \gamma_{f_\theta, p}(X))^\top (p_0(X) - p(X)) \\
&= \nabla_{\theta_i} f_\theta(X)^\top (\gamma(X) - q_{f_\theta, p}(X))^\top (p_0(X) - p(X)) \\
&\quad + \frac{1}{2} \mathbb{E}[\nabla_{\pi\pi\phi} \ell(W; f_\theta(X), \bar{p}(X))[\nabla_{\theta_i} f_\theta(X), p_0(X) - p(X), p(X) - p_0(X)]]
\end{aligned}$$

for some convex combination $\bar{p}(X)$ of $p(X)$ and $p_0(X)$.

We begin by bounding the target expectation using Cauchy-Schwarz

$$\begin{aligned}
&\mathbb{E}[\|\mathbf{n}(W; \theta, p, \gamma)\|_2^2] \\
&= \sum_{i \in [d]} \operatorname{Var}[\nabla_{\theta_i} \ell(W; f_\theta(X), p(X)) + \nabla_{\theta_i} f_\theta(X)^\top \gamma(X)^\top (Y - p(X))] \\
&= \sum_{i \in [d]} \operatorname{Var}[\nabla_{\theta_i} \ell(W; f_\theta(X), p_0(X)) + \Delta_i] \\
&= \sigma_0(\theta, p_0)^2 + \sum_{i \in [d]} \operatorname{Var}[\Delta_i] + 2\operatorname{Cov}(\nabla_{\theta_i} \ell(W; f_\theta(X), p_0(X)), \Delta_i) \\
&\leq \sigma_0(\theta, p_0)^2 + \sum_{i \in [d]} \operatorname{Var}[\Delta_i] + 2\sqrt{\operatorname{Var}[\nabla_{\theta_i} \ell(W; f_\theta(X), p_0(X))] \operatorname{Var}[\Delta_i]} \\
&\leq \sigma_0(\theta, p_0)^2 + (\sum_{i \in [d]} \operatorname{Var}[\Delta_i]) + 2\sqrt{\sum_{i \in [d]} \operatorname{Var}[\nabla_{\theta_i} \ell(W; f_\theta(X), p_0(X))] \sum_{i \in [d]} \operatorname{Var}[\Delta_i]} \\
&= (\sigma_0(\theta, p_0) + \sqrt{\sum_{i \in [d]} \operatorname{Var}[\Delta_i]})^2.
\end{aligned}$$

We next employ the law of total variance to rewrite the variance terms:

$$\begin{aligned}
\sum_{i \in [d]} \operatorname{Var}[\Delta_i] &= \sum_{i \in [d]} \operatorname{Var}[Z_i + \nabla_{\theta_i} f_\theta(X)^\top \gamma(X)^\top (Y - p_0(X))] \\
&= \mathbb{E}[\|\nabla_\theta f_\theta(X)^\top \gamma(X)^\top (Y - p_0(X))\|_2^2] + \sum_{i \in [d]} \operatorname{Var}[Z_i].
\end{aligned}$$

Finally, we control $\operatorname{Var}[Z_i]$ using Cauchy-Schwarz

$$\sqrt{\sum_{i \in [d]} \operatorname{Var}[Z_i]} \leq \|\nabla_\theta f_\theta^\top (\gamma_{f_\theta, p} - \gamma)^\top (p - p_0)\|_{2,2}.$$

$\square$

The two claims of Thm. 6 now follow from Theorems 2 and 3 of Ajalloeian & Stich (2020) respectively, with the parameters $\sigma^2$ and $\zeta$ instantiated with quantities $\sigma^2(\gamma)$ and $\zeta(\gamma)$ of Lemmas 7 and 8.

# I  EXPERIMENT DETAILS AND ADDITIONAL RESULTS

## I.1  TABULAR DATA

We use cross-fitting with 10 folds. The student is trained using the SEL loss with clipped teacher class probabilities $\max(\hat{p}(x), \epsilon)$ for $\epsilon = 10^{-3}$. The $\alpha$ hyperparameter of the loss correction was chosen by cross-validation with 5 folds. We repeat the experiments 5 times to measure the mean and standard deviation.

For the overfitting experiment, we use a random forest with 500 trees as the teacher and a random forest with 1-40 trees as the student.

We also evaluate the impact of teacher underfitting by limiting the teacher's maximum tree depth (from 1 to 20). Lower depth corresponds to greater underfitting. The teacher has 100 trees, and the student has 10 trees. For all of the datasets, loss correction successfully mitigates the teacher's underfitting and thus improves the student's performance. The effect is most pronounced when the teacher underfits more heavily (has lower tree depth).

We show the full results for all 5 of the datasets in Figs. 4 and 5.

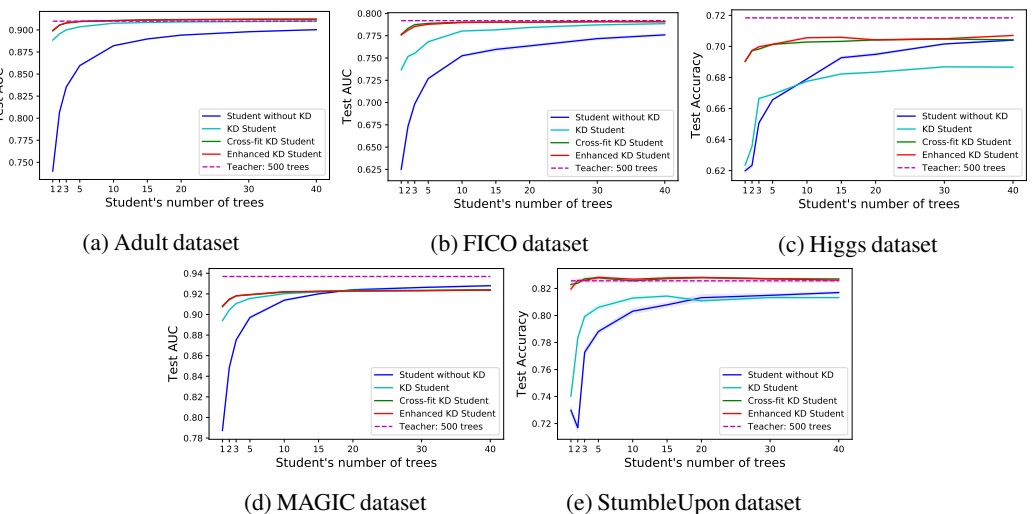

Figure 4: Tabular random forest distillation with varying student complexity.

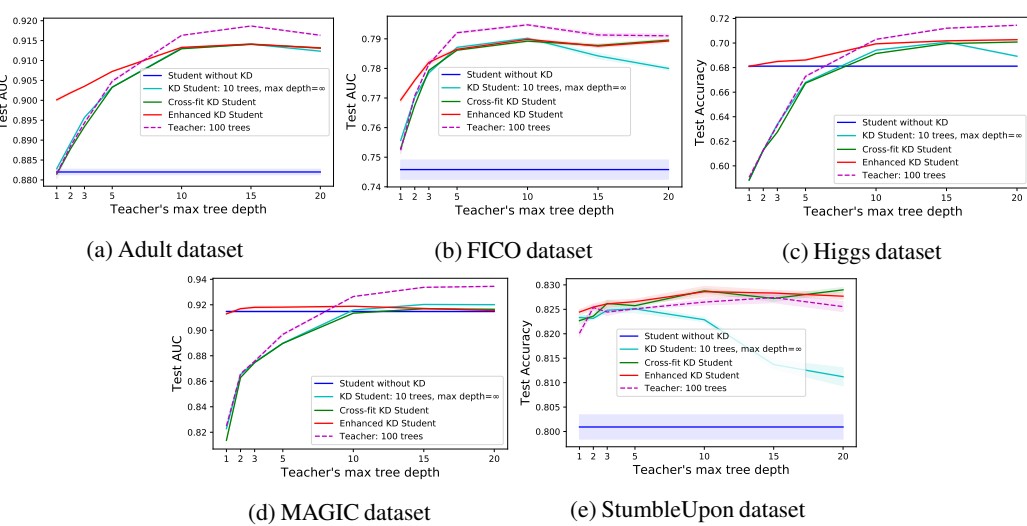

Figure 5: Tabular random forest distillation with varying teacher complexity.

## I.2   IMAGE DATA (CIFAR-10)

We use SGD with initial learning rate 0.1, momentum 0.9, and batch size 128 to train for 200 epochs. We use the standard learning rate decay schedule, where the learning rate is divided by 5 at epoch 60, 120, and 160. For loss correction, we select the value of the hyperparameter $\alpha$ that yields the highest accuracy on a held-out validation set. For cross-fitting, we use 10 folds.

