# OpenReview forum: "Knowledge Distillation as Semiparametric Inference"
_ICLR.cc/2021/Conference — ICLR 2021 Poster_

### Official Review · AnonReviewer3 · 2020-10-25
**Although viewing knowledge distillation as a semi-parametric inference problem is interesting, the related work of this paper is not sufficient and the key contribution of balancing bias and variance requires more effects.**

**Rating:** 6
**Confidence:** 4

**Review:**

This paper formulates knowledge distillation as a semi-parametric inference problem. Then, the paper adapts techniques from semi-parametric inference to analyze the error of a student model and improve the performance of knowledge distillation. The presentation is overall of good quality and it is relatively easy for me to follow the logic flow. That being said, there is still some room of improving the clarity of the paper.

P1: Analyzing and improving knowledge distillation from a point view of semi-parametric inference is very interesting.  The adaptation of semi-parametric inference techniques such as cross-fitting and orthogonal machine learning is novel.

C1: The related work is not sufficient. After reviewing some pioneer studies in knowledge distillation before 2015, this paper jumps to very recent studies in 2020.  As knowledge distillation is a hot research topic, there are a large number of related studies from 2016 to 2019. It is highly recommended for this paper to present the advance of knowledge distillation techniques from 2015 until now in a more comprehensive and organized way.

C2: Some contributions of this paper, cross-fitting and orthogonal machine learning, are largely adapted from semi-parametric inference literature. The contribution that is not adapted from semi-parametric inference literature is the proposed correction that balances the bias and variance. The correction is a linear combination of a bias term and a variance term. It is not clear how the bias and variance terms are related to the theoretical analysis in previous sections. It would be better if this paper could provide a clearer justification of the bias and variance terms by the theoretical analysis.

C3: The hyper-parameter \alpha is introduced to balance the bias and variance of knowledge distillation in the experiment section. Since one of the key ideas of the paper is to balance the bias and variance, the hyper-parameter should be important in determining the accuracy of the student model. This paper should do some ablation studies to investigate how the hyper-parameter affects the performance of the proposed knowledge distillation.

---

> ### Author Response · Authors · 2020-11-25
> **Response to Reviewer #3**
>
> We are encouraged that the reviewer finds the paper interesting and novel. We appreciate the reviewer’s constructive feedback.
>
> [Expanded related work] We have expanded our discussion of KD related work in the following ways: (1) we devote the end of Sec. 1 to reviewing theoretical advances in the understanding of KD and highlight every study of which we are aware that establishes theory for KD; (2) we summarize relevant empirical studies of why KD works and applications of KD in the extended literature review in App. A; (3) since we cannot review the vast literature on KD in its entirety,
> we point the interested reader to \citet{gou2020knowledge} for a recent overview of the field with an extensive bibliography. Please let us know if we have missed any other relevant references.
>
> [How gamma-corrected objective is derived from bias-variance tradeoff theory] In the revision (Sec. 5), we have added more details on how the gamma-corrected objective is derived from the bounds in Theorems 3 and 4. Thank you for the suggestion.
>
> [Ablation experiment on trading off bias and variance] We have added an experiment on varying the hyperparameter alpha that controls the tradeoff between bias and variance. When the variance is not controlled (i.e., Neyman orthogonal loss), the performance is even worse than training from scratch. When the bias is not controlled, the student does not benefit from the teacher. However, for intermediate values of alpha, this does indeed bring performance improvement.

---

### Official Review · AnonReviewer4 · 2020-10-27
**Review of Paper 2515**

**Rating:** 8
**Confidence:** 4

**Review:**

The knowledge distillation (KD) approach is a two-step procedure: first train the teacher model on the labeled data and then train the student model using the predicted class probabilities from the teacher model. A key theoretical question about KD is whether and how much this two-step approach can improve on the one-step approach that trains the student model directly on the labeled data. This paper casts KD as a semiparametric inference problem by treating the optimal student model as the parameter of primary interest and the true class probabilities as the nuisance parameter. Building on the semiparametric framework, the paper makes two contributions: 1) develops theoretical guarantees for the vanilla KD algorithm; 2) proposes improved KD by using a first-order bias-corrected loss and a sample splitting procedure.

Overall, I find the paper novel, well-written, and thought-provoking. It bridges the two directions of KD and semiparametric inference, allowing for the possibility of borrowing theoretical and methodological tools from semiparametric inference to analyze and improve the KD approach.

On the other hand, I think the paper somewhat oversells the semiparametric inference idea, since the theory presented in the paper (Theorem 1) does not have much semiparametric flavor. The central questions in semiparametric inference are the information bounds and semiparametric efficiency for the target parameter. Although Theorem 1 is useful, it does not directly address these key questions. For example, what is the best possible performance for the student model? In many classical semiparametric inference problems, we are able to construct a semiparametrically efficient estimator. Would this be possible for KD?

More comments:
1) The paper seems to treat $f_0$ as infinite dimensional (page 3, 3rd paragraph of Section 3). However, classical semiparametric inference largely exploits the finite or low dimensionality of the target parameter to answer the questions mentioned above. In the KD framework, I still think the finite-dimensional case is more interesting and would provide more fundamental insights.

2) There is a notable difference between the settings of KD and classical semiparametric inference: the target parameter $f_0$ in KD is ancillary in the sense that the data generating model depends only on the nuisance parameter $p_0$. Would this affect the applicability of semiparametric inference techniques?

3) Page 4, line 11, “student” should be “teacher”.

Update: I appreciate the authors’ response. The answer to my semiparametric efficiency question, however, is too terse and unclear. Theorems 1 and 3 provide only error bounds; I don’t see how they, together with the semiparametric efficiency of OLS, directly imply semiparametric efficiency for the student model. Semiparametric efficiency involves the optimal asymptotic variance. The authors might have confused the concept with rate optimality. Nevertheless, the paper is a nice contribution, and I will keep my rating.

---

> ### Author Response · Authors · 2020-11-25
> **Response to Reviewer #4**
>
> We appreciate the reviewer’s positive and constructive comments about our work.
>
> [Theory for finite-dimensional model class] Yes, our results can be used to construct semiparametrically efficient estimators for KD.  Here is a concrete example that we can include in the final revision.  Consider the squared error logit loss (1) and let F be a linear model class so that $f_0(x)$ is the best linear approximation to $\log(p_0(x))$ in a least squares sense.  Then, by Thm. 4 in our submission (now Thm. 3 in the revision), whenever the teacher MSE $\| p_0 - \hat{p} \|_{2,2} = o(n^{-1/4})$, our gamma-corrected cross-fitted student is semiparametrically efficient (because ordinary least squares is semiparametrically efficient for the linear projection model by Chamberlain “Asymptotic efficiency in estimation with conditional moment restrictions,” 1987).  Our Thm. 1 also implies semiparametric efficiency for vanilla distillation under the more stringent condition that the teacher MSE $\| p_0 - \hat{p} \|_{2,2} = o(n^{-1/2})$.
>
> [How ancillarity of $f_0$ affects applicability of semiparametric techniques] Thank you for this intriguing question. Our results do successfully adapt classical semiparametric inference techniques to the misspecified setting in which the true data generating process does not depend on $f_0$, and $f_0$ is only defined via its role as a population loss minimizer.  (See the “Theory for finite-dimensional model class” response above for a concrete example.)  The results of Foster and Syrgkanis (2019) on which we build also apply in this misspecified setting.  We will highlight these extensions beyond the classical semiparametric setting in the final revision.

---

### Official Review · AnonReviewer1 · 2020-10-30
**Review of "KNOWLEDGE DISTILLATION AS SEMIPARAMETRIC INFERENCE"**

**Rating:** 6
**Confidence:** 2

**Review:**

The paper under review presents a new approach to the theoretical understanding of konowledge distillation, an efficient technique that makes possible the training of small models given the predictions of a more complex model. The problem is recast as a problem in semi-supervised learning and several theorems are then drawn from this new viewpoint.

Based on this re-interpretation of the problem, a nice analysis of vanilla knowledge distillation is given and a correction is derived from a Taylor expansion of the loss which is then carefully investigated as well. Finally a stochastic gradient method is studied using relevant bias and variance parameters.

All the proofs seem correct. The simulations results provide convincing illustrations of the findings.

The exposition follows a fast pace and is not very easy to comprehend for the non-expert. In particular a more detailed presentation of the semi-supervised learning background should be given in order to better substantiate the subsequent findings of this paper.

---

> ### Author Response · Authors · 2020-11-25
> **Response to Reviewer #1**
>
> We thank the reviewer for their helpful feedback on our work. We are glad that the reviewer finds the experiments convincing illustrations for the theory.
> We have improved the clarity of exposition and provided more background to better explain the context of our theoretical results.

---

### Official Review · AnonReviewer2 · 2020-11-03
**Interesting approach for guarantees for KD. Final message muddied by lack of clarity and upselling of results.**

**Rating:** 6
**Confidence:** 4

**Review:**

### Summary
This paper gives generalization guarantees for vanilla knowledge distillation, where it identifies a large variance issue due to the teacher’s complexity in this guarantee, and then proposes alternative approaches to eliminate this issue. The framework it uses is inspired by semi-parametric methods. The main resulting algorithm has two key components: cross-fitting and a corrective surrogate loss. An SGD variant is also presented. Experiments illustrate some of the benefits of the cross-fitting approach.

### Strengths
+ The semi-parametric perspective and the attempt to have a bias-variance tradeoff as part of the knowledge distillation procedure are well-motivated and insightful.
+  Lemma 2 is a neat technical building block and it has the potential of being useful beyond the scope of the paper.

### Weaknesses
- The paper lacks clarity in notation and presentation. In terms of notation, many expressions are overloaded (multiple forms of $\gamma$ and completely different meanings for symbols $\zeta$ and $\mu$). In terms of clarity: important definitions are buried in the text instead of being highlighted (e.g. the corrective surrogate loss or the choice of, the definition of $q_p$ given only in an example), key steps are not always spelled out (e.g. the Bregman divergence simplifications, the choice of $\hat \gamma= q_{\hat p}$ in the cross-fitting approach, or the use of the teacher in the SGD approach)
- The motivation of the intermediate choices are not well-presented. For example, in transitioning to the corrective surrogate loss, a Taylor expansion of the ideal loss is used, but then the $\gamma$-corrected loss is given without explanation. To me it seems to be motivated only by quadratic losses, but I’m sure more can be said by the authors.
- Some of the arguments in favor of the paper are a bit underhanded. ~~For example, when discussing the drawbacks of vanilla distillation, the paper calls them *inherent*, though the only evidence is the upper bound provided. To make such a claim, a lower bound is needed.~~ [Edit: the author's lower bound is good.]
- More importantly, the key contribution of the approach, the claim of eliminating the teacher complexity from the upper bounds is not quite crisp. Namely, Theorem 4 requires a upper bound on the critical radii of *uniformly over all* classes that could be empirically selected by the teacher. It’s true that the complexity of the union of these classes (as in Theorem 1) can still be larger than this uniform bound. But this is not the comparison that the paper makes when it expounds the advantage of the new bound. Instead, it compares the complexity of any *one* of these classes (paragraph at the end of Section 4.2, page 6) to the union, which is disingenuous.
- Lastly, even though the corrective surrogate loss is touted as the key component of the approach, the experiments clearly show that it’s the cross-fitting that is ultimately regularizing the problem and reducing the dependence on the teacher’s complexity. But this isn’t analyzed in isolation in the paper. In general, the corrective loss only contributes modestly. The only exception seems in the Adult Dataset, but only when the teacher has *low* complexity, which is also not part of the message of the paper.

### Overall
I found that the paper takes an interesting approach to giving guarantees to KD methods, with several interesting technical contributions. However, the final message of the paper is muddied by the lack of clarity and an attempt to upsell the actual technical results. This is why at this point I would place the paper as borderline.

_[Edit: I read the response of the authors. The lower bound for vanilla KD seems good. But the author still fail to adequately explain the gain of Thm 4 (now Thm 3). The requisite upper bound on the critical radius uniformly over all potential functions generated by the teacher means that the result of the theorem still has an implicit dependence on the teacher's complexity. For this reason, I will leave my recommendation the same.]_

---

> ### Author Response · Authors · 2020-11-25
> **Response to Reviewer #2, part 2**
>
> [How teacher complexity can be eliminated from upper bounds] We apologize for the confusion concerning the statement of Thm 4 (in the original submission; now Thm 3 in the revision) and clarify here why the complexity quantity in Thm 4 is much smaller than that of Thm 1 and indeed does not suffer from the “statistical complexity” of the teacher’s function class. The quantity we need in Thm 1 is essentially a uniform bound only over the student’s function class for any fixed function of the teacher. For instance, if the loss function is L-lipschitz for any fixed hat{p} in the teacher’s function space, then by standard contraction theorems for Rademacher complexities, the critical radius of the class in Thm 1 is at most L times the critical radius of the student’s class. This more clearly shows that this quantity is only a statement about the statistical size of the student model and the teacher’s model enters only in a benign manner, since it is a statement for any fixed teacher and not uniformly over the teacher. In the paragraph following the theorem we provide another example in terms of VC dimension. We note that if the student and teacher function spaces have bounded VC dimension, then the quantity in Thm 4 can be upper bounded in terms of only the students VC dimension, while that in Thm 1 scales with the maximum of the VC dimensions of the student and the teacher. We will add more concrete comparison examples along these lines.
>
> [Analysis of cross-fitting alone] We will clarify that by plugging the constant choice $\hat{\gamma}^{(t)}(x) = 0$ into Thm. 4 of the original submission (now numbered Thm. 3 in the revision), one obtains an analysis of cross-fitting without gamma correction.  All of the gains surrounding the reduced dependence on the teacher’s model class still apply (i.e., Thm. 4 shows that cross-fitting alone is sufficient to reduce student error due to teacher overfitting).  However, cross-fitting without gamma correction is still susceptible to excessive student error due to teacher underfitting (the Thm. 4 error bound scales like $\|p_0 - \hat{p}\|^2$ instead of  $\|p_0 - \hat{p}\|^4$).  These qualitative predictions accord with our experimental observations as well.  For example, in Fig. 5 in the revision appendix, we see large gains from gamma correction (on top of cross-fitting) on the Higgs, MAGIC, and Adult datasets when the teacher is underfitting due to a restricted or misspecified function class, while the impact of cross-fitting is dominant when the teacher is more flexible and susceptible to overfitting.  We will endeavor to make this messaging more clear in the final revision.

---

> ### Author Response · Authors · 2020-11-25
> **Response to Reviewer #2, part 1**
>
> We thank the reviewer for their encouraging feedback and thoughtful comments, which have helped us improve our draft.
>
> [Motivation for the gamma-corrected loss] We have reorganized Sec. 4 to better motivate our gamma-corrected loss, and we would certainly appreciate any additional suggestions for improved clarity.  The gamma correction is developed in four steps: (1) we view the vanilla KD loss as a zero-th order estimate of the ideal loss and formulate a potentially more accurate ideal first-order estimate; (2) we approximate the ideal first-order estimate with an unbiased approximation of the unknown $p_0$ that we call the orthogonal loss; (3) we observe that for standard distillation losses (and more generally all Bregman divergence losses), the orthogonal loss is linear in f and reweights f by a certain matrix valued function $q_{f,p}$; and (4) we recognize that $q_{f,p}$ can have large variance for our target distillation losses due to small $p_0$ probabilities appearing in the denominator, so we replace $q_{f,p}$ with a tunable matrix function $\gamma$ that the user can choose to balance bias (when $\gamma = 0$ we recover the zero-th order approximation of vanilla KD) and variance (which is potentially large when $\gamma = q_{f,p}$).
>
> [“Inherent” drawbacks and lower bounds on vanilla KD’s performance] Thank you for highlighting this inappropriate language.  We agree that “inherent” is not quite the right way to describe the drawbacks of vanilla distillation, and we have updated the text to provide a more precise picture of vanilla distillation performance.  In addition, we have added two lower bounding examples to Sec. 3 in the revision, showing that the student’s performance does indeed depend on the teacher’s critical radius and approximation error.
>
> To demonstrate the worst-case dependence of vanilla distillation on the overfitting of the teacher to its training data, our first example instantiates a classification problem in which the following properties all hold simultaneously:
>
> The critical radius of the teacher-student function class $\mathcal{G}$ defined in Thm. 1 is a non-vanishing constant, due to the complexity of the teacher’s function class (the teacher is a truncated interpolation rule).
> The vanilla distillation student error is lower bounded by a non-vanishing constant $\| \hat{f} - f_0 \|^2$, matching the $\delta_n$ dependence of the Thm. 1 upper bound up to a constant factor.
> The cross-fitted distilled student in the same example satisfies $E \| \hat{f}^{CF} - f_0 \|_{2,2}^2 = O(n^{-4/(4+d)})$.
> In this example, the vanilla distillation student does not converge to the optimal $f_0$ due to the teacher’s overfitting, while the cross-fitted student will converge in MSE at a known rate.
>
> To demonstrate the worst-case dependence of vanilla distillation on the underfitting of the teacher due to large approximation error, our second example instantiates a classification problem in which the following properties all hold simultaneously:
>
> The vanilla distillation student error $\| \hat{f} - f_0 \|^2$ is lower bounded by $\| \gamma_{f_0,p_0}^\top (\hat{p} - p_0) \|^2$ with high probability, matching the teacher MSE dependence of the Thm. 1 upper bound up to a constant factor.
> Vanilla distillation student error $\| \hat{f} - f_0 \|^2$ converges at a slow $\Omega(min(1,\lambda^2))$ rate whenever the teacher regularization parameter $\lambda \geq \sqrt{\frac{C\log \log(n)}{n}}$.
> By Thm. 3 in the revision (Thm. 4 in the original submission), the gamma-corrected student in the same example satisfies $\| \hat{f} - f_0 \|^2 = O(\lambda^4 + 1/n)$ when $\lambda \geq \sqrt{\frac{C\log \log(n)}{n}}$.
>
> We also note that these examples describe the worst-case behavior of vanilla distillation and that in other better-case scenarios, vanilla distillation can perform better than the upper-bounding Thm. 1 would imply.
>
> Finally, we note that for the case of square losses, Section 5 of Foster and Syrgkanis, when instantiated to our setting with the square loss, provide conditions on the metric entropy of the student and teacher such that the orthogonal distillation approach (with gamma being equal to gamma_{f,p}), achieves minimax optimal oracle rates (in terms of dependence on n and the metric entropy) for the student’s excess risk problem (i.e. achieves the statistically optimal rate for the loss (log(p_0(x)) - f(x))^2, even if we had been given oracle access to the function p_0). We will add a remark on such oracle rates achieved by orthogonal distillation.

---

### Author Response · Authors · 2020-11-25
**Shared response**

We thank all the reviewers for their thoughtful feedback. We address general comments and questions from the reviewers here and then answer specific questions in individual responses. We have also uploaded a revised draft improving clarity in response to the reviewers’ suggestions and feedback.

[Cross-fitting and gamma correction] In the revision (see in particular Sec. 3), we have endeavored to make clearer that we introduce our two distinct KD enhancements (cross-fitting and gamma correction) to address two distinct KD failure modes.  Cross-fitting reduces student error due to data reuse and teacher overfitting to its training set, while gamma correction reduces student error due to teacher underfitting (i.e., when the teacher has a large approximation error due to an overly restrictive or misspecified function class).

[Improved clarity] We have improved our notation so that $\gamma_{f, p}$ is always used to represent the ground-truth gradient parameter and $\hat{\gamma}$ is always used for its estimate. We have also collected all the notations and definitions in a glossary in the appendix for easier reference.

[Added motivation for the gamma-corrected objective] In Sec. 5 of the revision, we have added more details on how the gamma-corrected objective arises from the bounds in Theorems 3 and 4.

---

### Decision · Program_Chairs · 2021-01-07
**Final Decision**

**Decision:**

Accept (Poster)

**Comment:**

The paper studies knowledge distillation through the lens of semi-parametric inference. There has been a lot of work on knowledge distillation in the past, but, as the paper points out, most of it is heuristic or empirical in nature, and thus theoretical understanding on the subject is lacking. The reviewers generally agree that this paper makes a useful theoretical contribution towards a better understanding of knowledge distillation. However, the reviewers also raised some concerns, especially regarding the clarity of the text, and they felt that the paper might be overstating its contribution. Still, all reviewers recommend acceptance, and on balance the merits of the paper seem to outweigh the weaknesses, so I'd be happy to recommend acceptance.